# iTRANSFORMER: INVERTED TRANSFORMERS ARE EFFECTIVE FOR TIME SERIES FORECASTING

**Yong Liu,**[*] **Tengge Hu,**[*] **Haoran Zhang,**[*] **Haixu Wu, Shiyu Wang**[§]**, Lintao Ma**[§]**, Mingsheng Long**[⊠]
School of Software, BNRist, Tsinghua University, Beijing 100084, China
[§]Ant Group, Hangzhou, China
{liuyong21,htg21,z-hr20,whx20}@mails.tsinghua.edu.cn
{weiming.wsy,lintao.mlt}@antgroup.com,mingsheng@tsinghua.edu.cn

## ABSTRACT

The recent boom of linear forecasting models questions the ongoing passion for architectural modifications of Transformer-based forecasters. These forecasters leverage Transformers to model the global dependencies over *temporal tokens* of time series, with each token formed by multiple variates of the same timestamp. However, Transformers are challenged in forecasting series with larger lookback windows due to performance degradation and computation explosion. Besides, the embedding for each temporal token fuses multiple variates that represent potential delayed events and distinct physical measurements, which may fail in learning variate-centric representations and result in meaningless attention maps. In this work, we reflect on the competent duties of Transformer components and repurpose the Transformer architecture without any modification to the basic components. We propose **iTransformer** that simply applies the attention and feed-forward network on the inverted dimensions. Specifically, the time points of individual series are embedded into *variate tokens* which are utilized by the attention mechanism to capture multivariate correlations; meanwhile, the feed-forward network is applied for each variate token to learn nonlinear representations. The iTransformer model achieves state-of-the-art on challenging real-world datasets, which further empowers the Transformer family with promoted performance, generalization ability across different variates, and better utilization of arbitrary lookback windows, making it a nice alternative as the fundamental backbone of time series forecasting. Code is available at this repository: https://github.com/thuml/iTransformer.

## 1 INTRODUCTION

Transformer (Vaswani et al., 2017) has achieved tremendous success in natural language processing (Brown et al., 2020) and computer vision (Dosovitskiy et al., 2021), growing into the foundation model that follows the scaling law (Kaplan et al., 2020). Inspired by the immense success in extensive fields, Transformer with strong capabilities of depicting pairwise dependencies and extracting multi-level representations in sequences is emerging in time series forecasting (Wu et al., 2021; Nie et al., 2023).

However, researchers have recently begun to question the validity of Transformer-based forecasters, which typically embed multiple variates of the same timestamp into indistinguishable channels and apply attention on these *temporal tokens* to capture temporal dependencies. Considering the numerical but less semantic relationship among time points, researchers find that simple linear layers, which can be traced back to statistical forecasters (Box & Jenkins, 1968), have exceeded complicated Transformers on both performance and efficiency (Zeng et al., 2023; Das et al., 2023). Meanwhile, ensuring the independence of variate and utilizing mutual

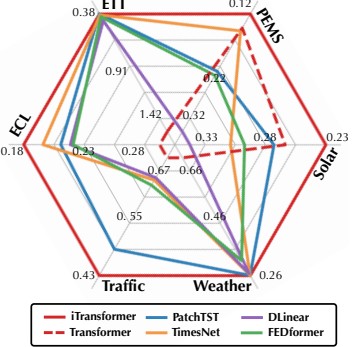

Figure 1: Performance of iTransformer. Average results (MSE) are reported following TimesNet (2023).

---

[*]Equal Contribution

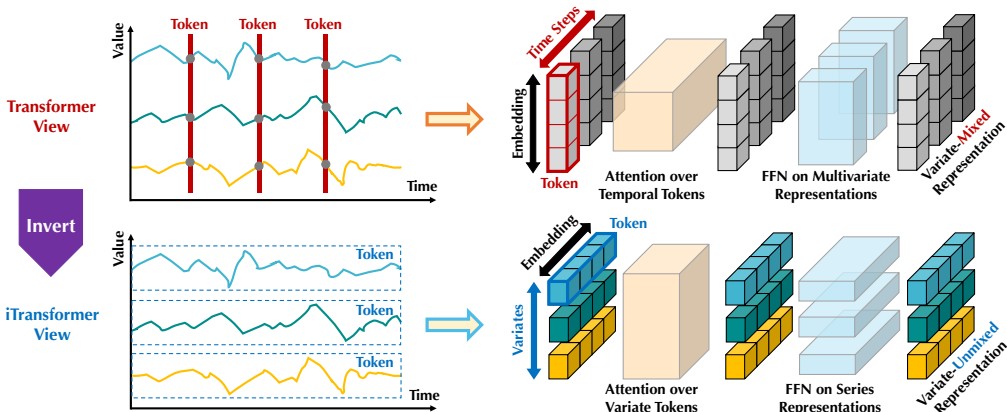

Figure 2: Comparison between the vanilla Transformer (top) and the proposed iTransformer (bottom). Transformer embeds the temporal token, which contains the multivariate representation of each time step. iTransformer embeds each series independently to the variate token, such that the attention module depicts the multivariate correlations and the feed-forward network encodes series representations.

information is ever more highlighted by recent research that explicitly models multivariate correlations to achieve accurate forecasting (Zhang & Yan, 2023; Ekambaram et al., 2023), but this goal can be hardly achieved without subverting the vanilla Transformer architecture.

Considering the disputes of Transformer-based forecasters, we reflect on why Transformers perform even worse than linear models in time series forecasting while acting predominantly in many other fields. We notice that the existing structure of Transformer-based forecasters may be not suitable for multivariate time series forecasting. As shown on the top of Figure 2, it is notable that the points of the same time step that basically represent completely different physical meanings recorded by inconsistent measurements are embedded into one token with wiped-out multivariate correlations. And the token formed by a single time step can struggle to reveal beneficial information due to excessively local receptive field and time-unaligned events represented by simultaneous time points. Besides, while series variations can be greatly influenced by the sequence order, permutation-invariant attention mechanisms are improperly adopted on the temporal dimension (Zeng et al., 2023). Consequently, Transformer is weakened to capture essential series representations and portray multivariate correlations, limiting its capacity and generalization ability on diverse time series data.

Concerning the potential risks of embedding multivariate points of a timestamp as a (temporal) token, we take an *inverted view* on time series and embed the whole time series of each variate independently into a (variate) token, the extreme case of Patching (Nie et al., 2023) that enlarges local receptive field. By inverting, the embedded token aggregates the global representations of series that can be more variate-centric and better leveraged by booming attention mechanisms for multivariate correlating. Meanwhile, the feed-forward network can be proficient enough to learn generalizable representations for distinct variates encoded from arbitrary lookback series and decoded to predict future series.

Based on the above motivations, we believe it is not that Transformer is ineffective for time series forecasting, but rather it is improperly used. In this paper, we revisit the structure of Transformer and advocate *iTransformer* as a fundamental backbone for time series forecasting. Technically, we embed each time series as *variate tokens*, adopt the attention for multivariate correlations, and employ the feed-forward network for series representations. Experimentally, the proposed iTransformer achieves state-of-the-art performance on real-world forecasting benchmarks shown in Figure 1 and surprisingly tackles the pain points of Transformer-based forecasters. Our contributions lie in three aspects:

- We reflect on the architecture of Transformer and refine that the competent capability of native Transformer components on multivariate time series is underexplored.

- We propose iTransformer that regards independent time series as tokens to capture multivariate correlations by self-attention and utilize layer normalization and feed-forward network modules to learn better series-global representations for time series forecasting.

- Experimentally, iTransformer achieves comprehensive state-of-the-art on real-world benchmarks. We extensively analyze the inverted modules and architecture choices, indicating a promising direction for the future improvement of Transformer-based forecasters.

## 2 RELATED WORK

With the progressive breakthrough made in natural language processing and computer vision areas, elaboratively designed Transformer variants are proposed to tackle ubiquitous time series forecasting applications. Going beyond contemporaneous TCNs (Bai et al., 2018; Liu et al., 2022a) and RNN-based forecasters (Zhao et al., 2017; Rangapuram et al., 2018; Salinas et al., 2020), Transformer has exhibited powerful sequence modeling capability and promising model scalability, leading to the trend of passionate modifications adapted for time series forecasting.

Through a systematical review of Transformer-based forecasters, we conclude that existing modifications can be divided into four categories by whether to modify the component and architecture. As shown in Figure 3, the first category (Wu et al., 2021; Li et al., 2021; Zhou et al., 2022), which is the most common practice, mainly concerns the component adaptation, especially the attention module for the temporal dependency modeling and the complexity optimization on long sequences. Nevertheless, with the rapid emergence of linear forecasters (Oreshkin et al., 2019; Zeng et al., 2023; Das et al., 2023; Liu et al., 2023), the impressive performance and efficiency continuously challenge this direction. Soon afterward, the second category attempts to fully utilize Transformer. It pays more attention to the inherent processing of time series, such as Stationarization (Liu et al., 2022b), Channel Independence, and Patching (Nie et al., 2023), which bring about consistently improved performance. Moreover, faced with the increasing significance of the independence and mutual interactions of multiple variates, the third category refurbishes Transformer in both aspects of component and architecture. Representative (Zhang & Yan, 2023) explicitly captures the cross-time and cross-variate dependencies by the renovated attention mechanism and architecture.

Unlike previous works, iTransformer modifies none of the native components of Transformer. Instead, we adopt the components on the inverted dimensions with the altered architecture, as the only one that belongs to the fourth category to our best knowledge. We believe the capabilities of the components have stood the test extensively, the truth is that the architecture of Transformer is improperly adopted.

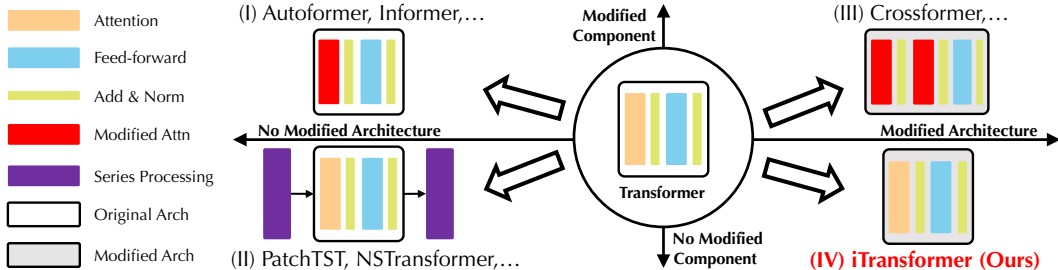

Figure 3: Transformer-based forecasters categorized by component and architecture modifications.

## 3 ITRANSFORMER

In multivariate time series forecasting, given historical observations $\mathbf{X} = \{\mathbf{x}_1, \ldots, \mathbf{x}_T\} \in \mathbb{R}^{T \times N}$ with $T$ time steps and $N$ variates, we predict the future $S$ time steps $\mathbf{Y} = \{\mathbf{x}_{T+1}, \ldots, \mathbf{x}_{T+S}\} \in \mathbb{R}^{S \times N}$. For convenience, we denote $\mathbf{X}_{t,:}$ as the simultaneously recorded time points at the step $t$, and $\mathbf{X}_{:,n}$ as the whole time series of each variate indexed by $n$. It is notable that $\mathbf{X}_{t,:}$ may not contain time points that essentially reflect the same event in real-world scenarios because of the systematical time lags among variates in the dataset. Besides, the elements of $\mathbf{X}_{t,:}$ can be distinct from each other in physical measurements and statistical distributions, for which a variate $\mathbf{X}_{:,n}$ generally shares.

### 3.1 STRUCTURE OVERVIEW

Our proposed *iTransformer* illustrated in Figure 4 adopts the *encoder-only* architecture of Transformer (Vaswani et al., 2017), including the embedding, projection, and Transformer blocks.

**Embedding the whole series as the token**   Most Transformer-based forecasters typically regard multiple variates of the same time as the (temporal) token and follow the generative formulation of forecasting tasks. However, we find the approach on the numerical modality can be less instructive for

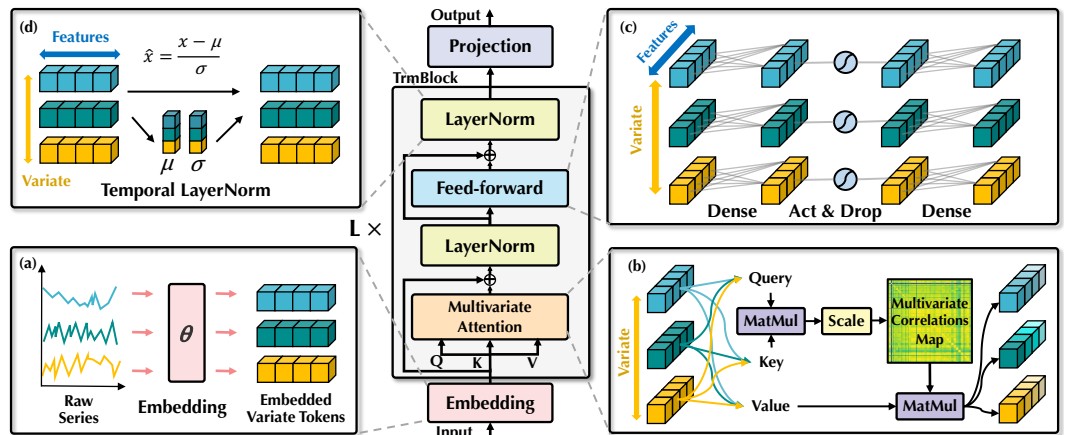

Figure 4: Overall structure of iTransformer, which shares the same modular arrangement with the encoder of Transformer. (a) Raw series of different variates are independently embedded as tokens. (b) Self-attention is applied to embedded variate tokens with enhanced interpretability revealing multivariate correlations. (c) Series representations of each token are extracted by the shared feed-forward network. (d) Layer normalization is adopted to reduce the discrepancies among variates.

learning attention maps, which is supported by increasing applications of Patching (Dosovitskiy et al., 2021; Nie et al., 2023) that broadens the respective field. Meanwhile, the triumph of linear forecasters also challenges the necessity of adopting a heavy encoder-decoder Transformer for generating tokens. Instead, our proposed encoder-only iTransformer focuses on representation learning and adaptive correlating of multivariate series. Each time series driven by the underlying complicated process is firstly tokenized to describe the properties of the variate, applied by self-attention for mutual interactions, and individually processed by feed-forward networks for series representations. Notably, the task to generate the predicted series is essentially delivered to linear layers, which has been proven competent by previous work (Das et al., 2023) and we provide a detailed analysis in the next section.

Based on the above considerations, in iTransformer, the process of predicting future series of each specific variate $\hat{\mathbf{Y}}_{:,n}$ based on the lookback series $\mathbf{X}_{:,n}$ is simply formulated as follows:

$$
\begin{aligned}
\mathbf{h}_n^0 &= \mathrm{Embedding}(\mathbf{X}_{:,n}), \\
\mathbf{H}^{l+1} &= \mathrm{TrmBlock}(\mathbf{H}^l),\ l = 0, \dots, L-1, \\
\hat{\mathbf{Y}}_{:,n} &= \mathrm{Projection}(\mathbf{h}_n^L),
\end{aligned}
\tag{1}
$$

where $\mathbf{H} = \{\mathbf{h}_1, \dots, \mathbf{h}_N\} \in \mathbb{R}^{N \times D}$ contains $N$ embedded tokens of dimension $D$ and the superscript denotes the layer index. Embedding $: \mathbb{R}^T \mapsto \mathbb{R}^D$ and Projection $: \mathbb{R}^D \mapsto \mathbb{R}^S$ are both implemented by multi-layer perceptron (MLP). The obtained variate tokens interact with each other by self-attention and are independently processed by the shared feed-forward network in each TrmBlock. Specifically, as the order of sequence is implicitly stored in the neuron permutation of the feed-forward network, the position embedding in the vanilla Transformer is no longer needed here.

**iTransformers** The architecture essentially presupposes no more specific requirements on Transformer variants, other than the attention is applicable for multivariate correlation. Thus, a bundle of efficient attention mechanisms (Li et al., 2021; Wu et al., 2022; Dao et al., 2022) can be the plugins, reducing the complexity when the variate number grows large. Besides, with the input flexibility of attention, the token number can vary from training to inference, and the model is allowed to be trained on arbitrary numbers of variates. The inverted Transformers, named *iTransformers*, are extensively evaluated in experiments of Section 4.2 and demonstrate advantages on time series forecasting.

## 3.2 INVERTED TRANSFORMER COMPONENTS

We organize a stack of $L$ blocks composed of the layer normalization, feed-forward network, and self-attention modules. But their duties on the inverted dimension are carefully reconsidered.

**Layer normalization**    Layer normalization (Ba et al., 2016) is originally proposed to increase the convergence and training stability of deep networks. In typical Transformer-based forecasters, the module normalizes the multivariate representation of the same timestamp, gradually fusing the variates with each other. Once the collected time points do not represent the same event, the operation will also introduce interaction noises between noncausal or delayed processes. In our inverted version, the normalization is applied to the series representation of individual variate as Equation 2, which has been studied and proved effective in tackling non-stationary problems (Kim et al., 2021; Liu et al., 2022b). Besides, since all series as (variate) tokens are normalized to a Gaussian distribution, the discrepancies caused by inconsistent measurements can be diminished. By contrast, in previous architecture, different tokens of time steps will be normalized, leading to oversmooth time series.

$$\text{LayerNorm}(\mathbf{H}) = \left\{ \left. \frac{\mathbf{h}_n - \text{Mean}(\mathbf{h}_n)}{\sqrt{\text{Var}(\mathbf{h}_n)}} \right| n = 1, \ldots, N \right\} \tag{2}$$

**Feed-forward network**    Transformer adopts the feed-forward network (FFN) as the basic building block for encoding token representation and it is identically applied to each token. As aforementioned, in the vanilla Transformer, multiple variates of the same timestamp that form the token can be malpositioned and too localized to reveal enough information for predictions. In the inverted version, FFN is leveraged on the series representation of each variate token. By the universal approximation theorem (Hornik, 1991), they can extract complicated representations to describe a time series. With the stacking of inverted blocks, they are devoted to encoding the observed time series and decoding the representations for future series using dense non-linear connections, which work effectively as the recent works completely built on MLPs (Tolstikhin et al., 2021; Das et al., 2023).

More interestingly, the identical linear operation on independent time series, which serves as the combination of the recent linear forecasters (Zeng et al., 2023) and Channel Independence (Nie et al., 2023), can be instructive for us to understand the series representations. Recent revisiting on linear forecasters (Li et al., 2023) highlights that temporal features extracted by MLPs are supposed to be shared within distinct time series. We propose a rational explanation that the neurons of MLP are taught to portray the intrinsic properties of any time series, such as the amplitude, periodicity, and even frequency spectrums (neuron as a filter), serving as a more advantageous predictive representation learner than the self-attention applied on time points. Experimentally, we validate that the division of labor helps enjoy the benefits of linear layers in Section 4.3, such as the promoted performance if providing enlarged lookback series, and the generalization ability on unseen variates.

**Self-attention**    While the attention mechanism is generally adopted for facilitating the temporal dependencies modeling in previous forecasters, the inverted model regards the whole series of one variate as an independent process. Concretely, with comprehensively extracted representations of each time series $\mathbf{H} = \{\mathbf{h}_0, \ldots, \mathbf{h}_N\} \in \mathbb{R}^{N \times D}$, the self-attention module adopts linear projections to get queries, keys, and values $\mathbf{Q}, \mathbf{K}, \mathbf{V} \in \mathbb{R}^{N \times d_k}$, where $d_k$ is the projected dimension.

With denotation of $\mathbf{q}_i, \mathbf{k}_j \in \mathbb{R}^{d_k}$ as the specific query and key of one (variate) token, we notice that each entry of the pre-Softmax scores is formulated as $\mathbf{A}_{i,j} = (\mathbf{QK}^\top / \sqrt{d_k})_{i,j} \propto \mathbf{q}_i^\top \mathbf{k}_j$. Since each token is previously normalized on its feature dimension, the entries can somewhat reveal the variate-wise correlation, and the whole score map $\mathbf{A} \in \mathbb{R}^{N \times N}$ exhibits the multivariate correlations between paired variate tokens. Consequently, highly correlated variate will be more weighted for the next representation interaction with values $\mathbf{V}$. Based on this intuition, the proposed mechanism is believed to be more natural and interpretable for multivariate series forecasting. We further provide the visualization analysis of the score map in Section 4.3 and Appendix E.1.

## 4    EXPERIMENTS

We thoroughly evaluate the proposed iTransformer on various time series forecasting applications, validate the generality of the proposed framework and further dive into the effectiveness of applying the Transformer components on the inverted dimensions of time series.

**Datasets**    We extensively include 7 real-world datasets in our experiments, including ECL, ETT (4 subsets), Exchange, Traffic, Weather used by Autoformer (Wu et al., 2021), Solar-Energy datasets

proposed in LSTNet (Lai et al., 2018), and PEMS (4 subsets) evaluated in SCINet (Liu et al., 2022a). We also provide the experiments on Market (6 subsets) in Appendix F.4. It records the minute-sampled server load of Alipay online transaction application with hundreds of variates, where we consistently outperform other baselines. Detailed dataset descriptions are provided in Appendix A.1.

## 4.1 FORECASTING RESULTS

In this section, we conduct extensive experiments to evaluate the forecasting performance of our proposed model together with advanced deep forecasters.

**Baselines** We carefully choose 10 well-acknowledged forecasting models as our benchmark, including (1) Transformer-based methods: Autoformer (Wu et al., 2021), FEDformer (Zhou et al., 2022), Stationary (Liu et al., 2022b), Crossformer (Zhang & Yan, 2023), PatchTST (Nie et al., 2023); (2) Linear-based methods: DLinear (Zeng et al., 2023), TiDE (Das et al., 2023), RLinear (Li et al., 2023); and (3) TCN-based methods: SCINet (Liu et al., 2022a), TimesNet (Wu et al., 2023).

**Main results** Comprehensive forecasting results are listed in Table 1 with the best in **red** and the second underlined. The lower MSE/MAE indicates the more accurate prediction result. Compared with other forecasters, iTransformer is particularly good at forecasting high-dimensional time series. Besides, PatchTST as the previous state-of-the-art, fails in many cases of PEMS, which can stem from the extremely fluctuating series of the dataset, and the patching mechanism of PatchTST may lose focus on specific locality to handle rapid fluctuation. By contrast, the proposed model aggregating the whole series variations for series representations can better cope with this situation. Notably, as the representative that explicitly captures multivariate correlations, the performance of Crossformer is still subpar to iTransformer, indicating the interaction of time-unaligned patches from different multivariate will bring about unnecessary noise for forecasting. Therefore, the native Transformer components are competent for temporal modeling and multivariate correlating, and the proposed inverted architecture can effectively tackle real-world time series forecasting scenarios.

Table 1: Multivariate forecasting results with prediction lengths $S \in \{12, 24, 36, 48\}$ for PEMS and $S \in \{96, 192, 336, 720\}$ for others and fixed lookback length $T = 96$. Results are averaged from all prediction lengths. *Avg* means further averaged by subsets. Full results are listed in Appendix F.4.

| Models | iTransformer (Ours) | | RLinear (2023) | | PatchTST (2023) | | Crossformer (2023) | | TiDE (2023) | | TimesNet (2023) | | DLinear (2023) | | SCINet (2022a) | | FEDformer (2022) | | Stationary (2022b) | | Autoformer (2021) | |
|---|---|---|---|---|---|---|---|---|---|---|---|---|---|---|---|---|---|---|---|---|---|---|
| Metric | MSE | MAE | MSE | MAE | MSE | MAE | MSE | MAE | MSE | MAE | MSE | MAE | MSE | MAE | MSE | MAE | MSE | MAE | MSE | MAE | MSE | MAE |
| ECL | **0.178** | **0.270** | 0.219 | 0.298 | 0.205 | 0.290 | 0.244 | 0.334 | 0.251 | 0.344 | 0.192 | 0.295 | 0.212 | 0.300 | 0.268 | 0.365 | 0.214 | 0.327 | 0.193 | 0.296 | 0.227 | 0.338 |
| ETT (Avg) | 0.383 | 0.399 | **0.380** | **0.392** | 0.381 | 0.397 | 0.685 | 0.578 | 0.482 | 0.470 | 0.391 | 0.404 | 0.442 | 0.444 | 0.689 | 0.597 | 0.408 | 0.428 | 0.471 | 0.464 | 0.465 | 0.459 |
| Exchange | 0.360 | **0.403** | 0.378 | 0.417 | 0.367 | 0.404 | 0.940 | 0.707 | 0.370 | 0.413 | 0.416 | 0.443 | **0.354** | 0.414 | 0.750 | 0.626 | 0.519 | 0.429 | 0.461 | 0.454 | 0.613 | 0.539 |
| Traffic | **0.428** | **0.282** | 0.626 | 0.378 | 0.481 | 0.304 | 0.550 | 0.304 | 0.760 | 0.473 | 0.620 | 0.336 | 0.625 | 0.383 | 0.804 | 0.509 | 0.610 | 0.376 | 0.624 | 0.340 | 0.628 | 0.379 |
| Weather | **0.258** | **0.278** | 0.272 | 0.291 | 0.259 | 0.281 | 0.259 | 0.315 | 0.271 | 0.320 | 0.259 | 0.287 | 0.265 | 0.317 | 0.292 | 0.363 | 0.309 | 0.360 | 0.288 | 0.314 | 0.338 | 0.382 |
| Solar-Energy | **0.233** | **0.262** | 0.369 | 0.356 | 0.270 | 0.307 | 0.641 | 0.639 | 0.347 | 0.417 | 0.301 | 0.319 | 0.330 | 0.401 | 0.282 | 0.375 | 0.291 | 0.381 | 0.261 | 0.381 | 0.885 | 0.711 |
| PEMS (Avg) | **0.119** | **0.218** | 0.514 | 0.482 | 0.217 | 0.305 | 0.220 | 0.304 | 0.375 | 0.440 | 0.148 | 0.246 | 0.320 | 0.394 | 0.121 | 0.222 | 0.224 | 0.327 | 0.151 | 0.249 | 0.614 | 0.575 |

## 4.2 ITRANSFORMERS GENERALITY

In this section, we evaluate *iTransformers* by applying our framework to Transformer and its variants, which generally address the quadratic complexity of the self-attention mechanism, including Reformer (Kitaev et al., 2020), Informer (Li et al., 2021), Flowformer (Wu et al., 2022) and FlashAttention (Dao et al., 2022). Surprising and promising discoveries are exhibited, indicating the simple inverted perspective can enhance Transformer-based forecasters with promoted performance with efficiency, generalization on unseen variates, and better utilization of historical observations.

**Performance promotion** We evaluate Transformers and the corresponding iTransformers with the reported performance promotions in Table 2. It is notable that the framework consistently improves various Transformers. Overall, it achieves averaged **38.9%** promotion on Transformer, **36.1%** on Reformer, **28.5%** on Informer, **16.8%** on Flowformer and **32.2%** on Flashformer, revealing the previous improper usage of the Transformer architecture on time series forecasting. Moreover, since the attention mechanism is adopted on the variate dimension in our inverted structure, the introduction of efficient attentions with linear complexity essentially addresses the computational problem due to

numerous variates, which is prevalent in real-world applications but can be resource-consuming for Channel Independence (Nie et al., 2023). Therefore, the idea of iTransformer can be widely practiced on Transformer-based forecasters to take advantage of booming efficient attention mechanisms.

Table 2: Performance promotion obtained by our inverted framework. Flashformer means Transformer equipped with hardware-accelerated FlashAttention (Dao et al., 2022). We report the average performance and the relative MSE reduction (Promotion). Full results can be found in Appendix F.2.

| Models | | Transformer (2017) | | Reformer (2020) | | Informer (2021) | | Flowformer (2022) | | Flashformer (2022) | |
|---|---|---|---|---|---|---|---|---|---|---|---|
| Metric | | MSE | MAE | MSE | MAE | MSE | MAE | MSE | MAE | MSE | MAE |
| ECL | Original | 0.277 | 0.372 | 0.338 | 0.422 | 0.311 | 0.397 | 0.267 | 0.359 | 0.285 | 0.377 |
| | **+Inverted** | **0.178** | **0.270** | **0.208** | **0.301** | **0.216** | **0.311** | **0.210** | **0.293** | **0.206** | **0.291** |
| | Promotion | 35.6% | 27.4% | 38.4% | 28.7% | 30.5% | 21.6% | 21.3% | 18.6% | 27.8% | 22.9% |
| Traffic | Original | 0.665 | 0.363 | 0.741 | 0.422 | 0.764 | 0.416 | 0.750 | 0.421 | 0.658 | 0.356 |
| | **+Inverted** | **0.428** | **0.282** | **0.647** | **0.370** | **0.662** | **0.380** | **0.524** | **0.355** | **0.492** | **0.333** |
| | Promotion | 35.6% | 22.3% | 12.7% | 12.3% | 13.3% | 8.6% | 30.1% | 15.6% | 25.2% | 6.4% |
| Weather | Original | 0.657 | 0.572 | 0.803 | 0.656 | 0.634 | 0.548 | 0.286 | 0.308 | 0.659 | 0.574 |
| | **+Inverted** | **0.258** | **0.279** | **0.248** | **0.292** | **0.271** | **0.330** | **0.266** | **0.285** | **0.262** | **0.282** |
| | Promotion | 60.2% | 50.8% | 69.2% | 55.5% | 57.3% | 39.8% | 7.2% | 7.7% | 60.2% | 50.8% |

**Variate generalization**    By inverting vanilla Transformers, it is notable that the models are empowered with the generalization capability on unseen variates. Firstly, benefiting from the flexibility of the number of input tokens, the amount of variate channels is no longer restricted and thus feasible to vary from training and inference. Besides, feed-forward networks are identically applied on independent variate tokens in iTransformer. As aforementioned, the neurons as filters learn the intrinsic patterns of any time series, which are inclined to be shared and transferable among distinct variates.

To verify the hypothesis, we compare inverting with another generalizing strategy: Channel Independence, training a shared backbone to forecast all variates. We partition the variates of each dataset into five folders, train models with only 20% of variates of one folder, and directly forecast all variates without fine-tuning. We compare the performance in Figure 5 and each bar presents the averaged results of all folders to avoid the randomness of partition. CI-Transformers take a long time to predict each variate one by one during inference while iTransformers directly predict all variates and generally present smaller increases, indicating FFN is competent to learn transferable time series representations. It leaves a potential direction to build a foundation model upon iTransformer, where diverse multivariate time series with different numbers of variates can be feasibly trained together.

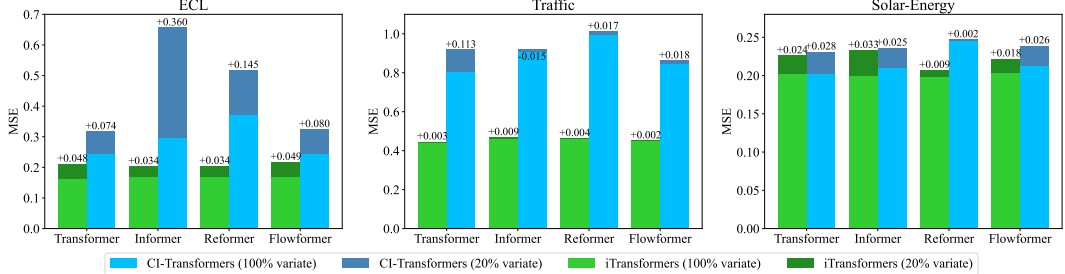

Figure 5: Performance of generalization on unseen variates. We partition the variates of each dataset into five folders, train models with 20% variates, and use the partially trained model to forecast all varieties. iTransformers can be trained efficiently and forecast with good generalizability.

**Increasing lookback length**    Previous works have witnessed the phenomenon that the forecasting performance does not necessarily improve with the increase of lookback length on Transformers (Nie et al., 2023; Zeng et al., 2023), which can be attributed to the distracted attention on the growing input. However, the desired performance improvement is generally held on linear forecasts, theoretically supported by statistical methods (Box & Jenkins, 1968) with enlarged historical information to be

utilized. As the working dimensions of attention and feed-forward network are inverted, we evaluate the performance of Transformers and iTransformer in Figure 6 with increased lookback length. The results surprisingly verify the rationality of leveraging MLPs on the temporal dimension such that Transformers can benefit from the extended lookback window for more precise predictions.

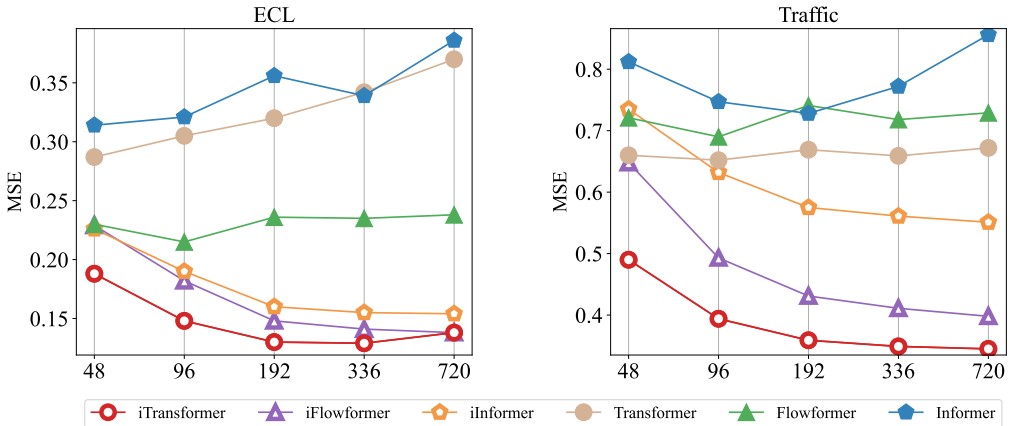

Figure 6: Forecasting performance with the lookback length $T \in \{48, 96, 192, 336, 720\}$ and fixed prediction length $S = 96$. While the performance of Transformer-based forecasters does not necessarily benefit from the increased lookback length, the inverted framework empowers the vanilla Transformer and its variants with improved performance on the enlarged lookback window.

## 4.3 MODEL ANALYSIS

**Ablation study** To verify the rational business of Transformer components, we provide detailed ablations covering both replacing components (Replace) and removing components (w/o) experiments. The results are listed in Table 3. iTransformer that utilizes attention on the variate dimension and feed-forward on the temporal dimension generally achieves the best performance. Notably, the performance of vanilla Transformer (the third row) performs the worst among these designs, revealing the potential risks of the conventional architecture, which we describe in detail in Appendix E.3.

Table 3: Ablations on iTransformer. We replace different components on the respective dimension to learn multivariate correlations (Variate) and series representations (Temporal), in addition to component removal. The average results of all predicted lengths are listed here.

| Design | Variate | Temporal | ECL | | Traffic | | Weather | | Solar-Energy | |
|---|---|---|---|---|---|---|---|---|---|---|
| | | | MSE | MAE | MSE | MAE | MSE | MAE | MSE | MAE |
| **iTransformer** | **Attention** | **FFN** | **0.178** | **0.270** | **0.428** | **0.282** | 0.258 | 0.278 | **0.233** | **0.262** |
| | Attention | Attention | 0.193 | 0.293 | 0.913 | 0.500 | 0.255 | 0.280 | 0.261 | 0.291 |
| Replace | FFN | Attention | 0.202 | 0.300 | 0.863 | 0.499 | 0.258 | 0.283 | 0.285 | 0.317 |
| | FFN | FFN | 0.182 | 0.287 | 0.599 | 0.348 | **0.248** | **0.274** | 0.269 | 0.287 |
| w/o | Attention | w/o | 0.189 | 0.278 | 0.456 | 0.306 | 0.261 | 0.281 | 0.258 | 0.289 |
| | w/o | FFN | 0.193 | 0.276 | 0.461 | 0.294 | 0.265 | 0.283 | 0.261 | 0.283 |

**Analysis of series representations** To further validate the claim that feed-forward networks are more favored to extract the series representations. We conduct representation analysis based on the centered kernel alignment (CKA) similarity (Kornblith et al., 2019). A higher CKA indicates more similar representations. For Transformer variants and iTransformers, we calculate the CKA between the output features of the first and the last block. Notably, previous works have demonstrated that time series forecasting, as a low-level generative task, prefers the higher CKA similarity (Wu et al., 2023; Dong et al., 2023) for the better performance. As shown in Figure 7, a clear division line is exhibited, implying that iTransformers have learned more appropriate series representations by inverting the dimension and thus achieve more accurate predictions. The results also advocate inverting Transformer deserves a fundamental renovation of the forecasting backbone.

**Analysis of multivariate correlations** By assigning the duty of multivariate correlation to the attention mechanism, the learned map enjoys enhanced interpretability. We present the case visualization on series from Solar-Energy in Figure 7, which has distinct correlations in the lookback and future windows. It can be observed that in the shallow attention layer, the learned map shares lots of similarities to the correlations of raw input series. As it dives into deeper layers, the learned map become gradually alike to the correlations of future series, which validates the inverted operation empowers interpretable attention for correlating, and the processes of encoding the past and decoding for the future are essentially conducted in series representations during feed-forwarding.

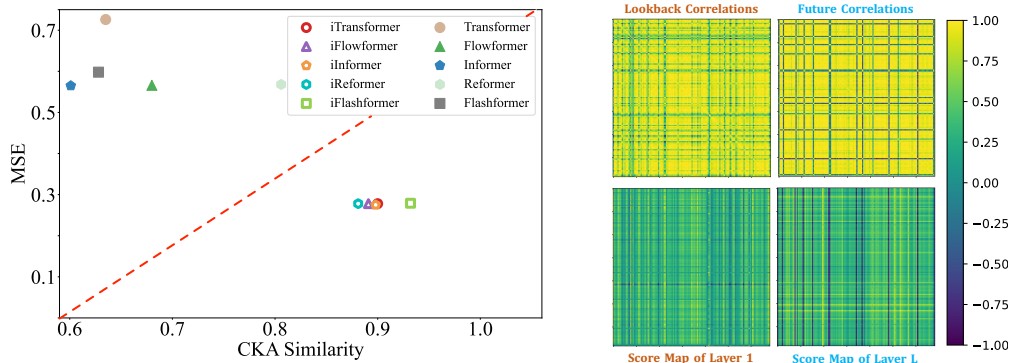

Figure 7: Analysis of series representations and multivariate correlations. Left: MSE and CKA similarity of representations comparison between Transformers and iTransformers. A higher CKA similarity indicates more favored representations for accurate predictions. Right: A case visualization of multivariate correlations of raw time series and the learned score maps by inverted self-attention.

**Efficient training strategy** Due to the quadratic complexity of self-attention, it can be overwhelming for training on numerous variates, which is very common in real-world scenarios. In addition to efficient attention mechanisms, we propose a novel training strategy for high-dimensional multivariate series by taking advantage of previously demonstrated variate generation capability. Concretely, we randomly choose part of the variates in each batch and only train the model with selected variates. Since the number of variate channels is flexible because of our inverting, the model can predict all the variates for predictions. As shown in Figure 8, the performance of our proposed strategy is still comparable with full-variate training, while the memory footprint can be reduced significantly.

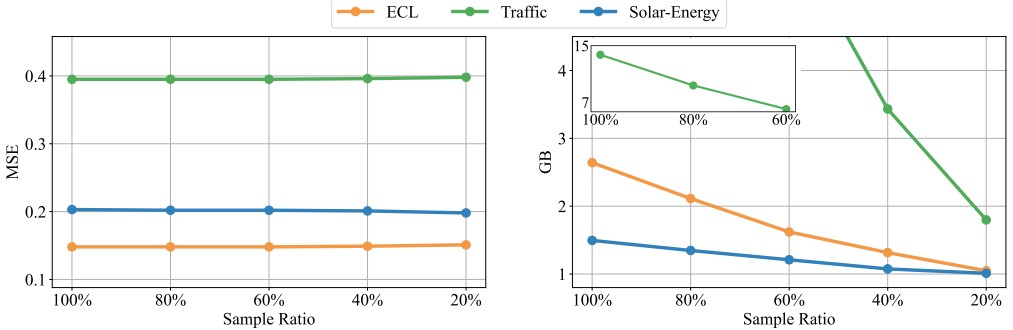

Figure 8: Analysis of the efficient training strategy. While the performance (left) remains stable on partially trained variates of each batch with different sampled ratios, the memory footprint (right) can be cut off greatly. We provide the comprehensive model efficiency analysis in Appendix D.

## 5 CONCLUSION AND FUTURE WORK

Considering the characteristics of multivariate time series, we propose iTransformer that inverts the structure of Transformer without modifying any native modules. iTransformer regards independent series as variate tokens to capture multivariate correlations by attention and utilize layer normalization and feed-forward networks to learn series representations. Experimentally, iTransformer achieves state-of-the-art performance and exhibits remarkable framework generality supported by promising analysis. In the future, we will explore large-scale pre-training and more time series analysis tasks.

## 6 ETHICS STATEMENT

Our work only focuses on the time series forecasting problem, so there is no potential ethical risk.

## 7 REPRODUCIBILITY STATEMENT

In the main text, we have strictly formalized the model architecture with equations. All the implementation details are included in the Appendix, including dataset descriptions, metrics, model, and experiment configurations. The code will be made public once the paper is accepted.

## ACKNOWLEDGMENTS

This work was supported by the National Key Research and Development Plan (2021YFB1715200), the National Natural Science Foundation of China (U2342217 and 62022050), the BNRist Innovation Fund (BNR2024RC01010), Ant Group through CCF-Ant Research Fund, and the National Engineering Research Center for Big Data Software.

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

# A IMPLEMENTATION DETAILS

## A.1 DATASET DESCRIPTIONS

We conduct experiments on 7 real-world datasets to evaluate the performance of the proposed iTransformer including (1) ETT (Li et al., 2021) contains 7 factors of electricity transformer from July 2016 to July 2018. There are four subsets where ETTh1 and ETTh2 are recorded every hour, and ETTm1 and ETTm2 are recorded every 15 minutes. (2) Exchange (Wu et al., 2021) collects the panel data of daily exchange rates from 8 countries from 1990 to 2016. (3) Weather (Wu et al., 2021) includes 21 meteorological factors collected every 10 minutes from the Weather Station of the Max Planck Biogeochemistry Institute in 2020. (4) ECL (Wu et al., 2021) records the hourly electricity consumption data of 321 clients. (5) Traffic (Wu et al., 2021) collects hourly road occupancy rates measured by 862 sensors of San Francisco Bay area freeways from January 2015 to December 2016. (6) Solar-Energy (Lai et al., 2018) records the solar power production of 137 PV plants in 2006, which are sampled every 10 minutes. (7) PEMS contains the public traffic network data in California collected by 5-minute windows. We use the same four public subsets (PEMS03, PEMS04, PEMS07, PEMS08) adopted in SCINet (Liu et al., 2022a).

Apart from the public datasets widely used as forecasting benchmarks, we also collect a set of Market datasets of a real-world application, which records the minute-sampled server load of Alipay online transactions between January 30th, 2023, and April 9th, 2023 with the number of variates varied from 285 to 759. It includes 6 sub-datasets, which are divided according to diverse transaction domains.

We follow the same data processing and train-validation-test set split protocol used in TimesNet (Wu et al., 2023), where the train, validation, and test datasets are strictly divided according to chronological order to make sure there are no data leakage issues. As for the forecasting settings, we fix the length of the lookback series as 96 in ETT, Weather, ECL, Solar-Energy, PEMS, and Traffic, and the prediction length varies in $\{96, 192, 336, 720\}$. For the PEMS dataset, the prediction length varies in $\{12, 24, 36, 48\}$, which is the same as SCINet, the previous state-of-the-art on this dataset. For the Market dataset, the lookback contains the past one day observations with $144$ time points and the forecasting length varies in $\{12, 24, 72, 144\}$. The details of datasets are provided in Table 4.

## A.2 IMPLEMENTATION DETAILS

---

**Algorithm 1** iTransformer - Overall Architecture.

---

**Require:** Input lookback time series $\mathbf{X} \in \mathbb{R}^{T \times N}$; input Length $T$; predicted length $S$; variates number $N$; token dimension $D$; iTransformer block number $L$.

1: $\mathbf{X} = \mathbf{X}.\texttt{transpose}$      $\triangleright \mathbf{X} \in \mathbb{R}^{N \times T}$

2: $\triangleright$ Multi-layer Perceptron works on the last dimension to embed series into variate tokens.

3: $\mathbf{H}^0 = \texttt{MLP}(\mathbf{X})$      $\triangleright \mathbf{H}^0 \in \mathbb{R}^{N \times D}$

4: **for** $l$ **in** $\{1, \ldots, L\}$**:**      $\triangleright$ Run through iTransformer blocks.

5:      $\triangleright$ Self-attention layer is applied on variate tokens.

6:      $\mathbf{H}^{l-1} = \texttt{LayerNorm}\big(\mathbf{H}^{l-1} + \texttt{Self-Attn}(\mathbf{H}^{l-1})\big)$      $\triangleright \mathbf{H}^{l-1} \in \mathbb{R}^{N \times D}$

7:      $\triangleright$ Feed-forward network is utilized for series representations, broadcasting to each token.

8:      $\mathbf{H}^l = \texttt{LayerNorm}\big(\mathbf{H}^{l-1} + \texttt{Feed-Forward}(\mathbf{H}^{l-1})\big)$      $\triangleright \mathbf{H}^l \in \mathbb{R}^{N \times D}$

9:      $\triangleright$ LayerNorm is adopted on series representations to reduce variates discrepancies.

10: **End for**

11: $\hat{\mathbf{Y}} = \texttt{MLP}(\mathbf{H}^L)$      $\triangleright$ Project tokens back to predicted series, $\hat{\mathbf{Y}} \in \mathbb{R}^{N \times S}$

12: $\hat{\mathbf{Y}} = \hat{\mathbf{Y}}.\texttt{transpose}$      $\triangleright \hat{\mathbf{Y}} \in \mathbb{R}^{S \times N}$

13: **Return** $\hat{\mathbf{Y}}$      $\triangleright$ Return the prediction result $\hat{\mathbf{Y}}$

---

Table 4: Detailed dataset descriptions. *Dim* denotes the variate number of each dataset. *Dataset Size* denotes the total number of time points in (Train, Validation, Test) split respectively. *Prediction Length* denotes the future time points to be predicted and four prediction settings are included in each dataset. *Frequency* denotes the sampling interval of time points.

| Dataset | Dim | Prediction Length | Dataset Size | Frequency | Information |
|---|---|---|---|---|---|
| ETTh1, ETTh2 | 7 | {96, 192, 336, 720} | (8545, 2881, 2881) | Hourly | Electricity |
| ETTm1, ETTm2 | 7 | {96, 192, 336, 720} | (34465, 11521, 11521) | 15min | Electricity |
| Exchange | 8 | {96, 192, 336, 720} | (5120, 665, 1422) | Daily | Economy |
| Weather | 21 | {96, 192, 336, 720} | (36792, 5271, 10540) | 10min | Weather |
| ECL | 321 | {96, 192, 336, 720} | (18317, 2633, 5261) | Hourly | Electricity |
| Traffic | 862 | {96, 192, 336, 720} | (12185, 1757, 3509) | Hourly | Transportation |
| Solar-Energy | 137 | {96, 192, 336, 720} | (36601, 5161, 10417) | 10min | Energy |
| PEMS03 | 358 | {12, 24, 48, 96} | (15617, 5135, 5135) | 5min | Transportation |
| PEMS04 | 307 | {12, 24, 48, 96} | (10172, 3375, 3375) | 5min | Transportation |
| PEMS07 | 883 | {12, 24, 48, 96} | (16911, 5622, 5622) | 5min | Transportation |
| PEMS08 | 170 | {12, 24, 48, 96} | (10690, 3548, 3548) | 5min | Transportation |
| Market-Merchant | 285 | {12, 24, 72, 144} | (7045, 1429, 1429) | 10min | Transaction |
| Market-Wealth | 485 | {12, 24, 72, 144} | (7045, 1429, 1429) | 10min | Transaction |
| Market-Finance | 405 | {12, 24, 72, 144} | (7045, 1429, 1429) | 10min | Transaction |
| Market-Terminal | 307 | {12, 24, 72, 144} | (7045, 1429, 1429) | 10min | Transaction |
| Market-Payment | 759 | {12, 24, 72, 144} | (7045, 1429, 1429) | 10min | Transaction |
| Market-Customer | 395 | {12, 24, 72, 144} | (7045, 1429, 1429) | 10min | Transaction |

All the experiments are implemented in PyTorch (Paszke et al., 2019) and conducted on a single NVIDIA P100 16GB GPU. We utilize ADAM (Kingma & Ba, 2015) with an initial learning rate in $\{10^{-3}, 5 \times 10^{-4}, 10^{-4}\}$ and L2 loss for the model optimization. The batch size is uniformly set to 32 and the number of training epochs is fixed to 10. We set the number of inverted Transformer blocks in our proposed model $L \in \{2, 3, 4\}$. The dimension of series representations $D$ is set from $\{256, 512\}$. All the compared baseline models that we reproduced are implemented based on the benchmark of TimesNet (Wu et al., 2023) Repository, which is fairly built on the configurations provided by each model's original paper or official code. We provide the pseudo-code of iTransformer in Algorithm 1. We also report the standard deviation of iTransformer performance under five runs with different random seeds in Table 5, which exhibits that the performance of iTransformer is stable.

Table 5: Robustness of iTransformer performance. The results are obtained from five random seeds.

| Dataset | | ECL | | ETTh2 | | Exchange | |
|---|---|---|---|---|---|---|---|
| Horizon | | MSE | MAE | MSE | MAE | MSE | MAE |
| 96 | | 0.148±0.000 | 0.240±0.000 | 0.297±0.002 | 0.349±0.001 | 0.088±0.001 | 0.209±0.001 |
| 192 | | 0.162±0.002 | 0.253±0.002 | 0.380±0.001 | 0.400±0.001 | 0.181±0.001 | 0.304±0.001 |
| 336 | | 0.178±0.000 | 0.269±0.001 | 0.428±0.002 | 0.432±0.001 | 0.334±0.001 | 0.419±0.001 |
| 720 | | 0.225±0.006 | 0.317±0.007 | 0.427±0.004 | 0.445±0.002 | 0.829±0.012 | 0.691±0.005 |

| Dataset | | Solar-Energy | | Traffic | | Weather | |
|---|---|---|---|---|---|---|---|
| Horizon | | MSE | MAE | MSE | MAE | MSE | MAE |
| 96 | | 0.203±0.002 | 0.237±0.002 | 0.395±0.001 | 0.268±0.001 | 0.174±0.000 | 0.214±0.000 |
| 192 | | 0.233±0.002 | 0.261±0.001 | 0.417±0.002 | 0.276±0.001 | 0.221±0.002 | 0.254±0.001 |
| 336 | | 0.248±0.000 | 0.273±0.000 | 0.433±0.004 | 0.283±0.000 | 0.278±0.002 | 0.296±0.001 |
| 720 | | 0.249±0.001 | 0.275±0.000 | 0.467±0.003 | 0.302±0.000 | 0.358±0.000 | 0.349±0.000 |

## B  ABLATION STUDIES

To elaborate on the rational business of Transformer components, we conduct detailed ablations covering replacing components (Replace) and removing components (w/o). Since the average results are listed in Table 3 due to the paper limit, we provide detailed results and analysis here.

As shown in Table 6, among various architectural designs, iTransformer generally exhibits superior performance, which learns multivariate correlations by self-attention and encodes series representations by FFN. Nevertheless, the arrangement of the vanilla Transformer can lead to degenerated performance, indicating the misuse of Transformer components on the time series modality. Based on the relatively poor results of the second (both attentions) and the third (the vanilla Transformer) designs, one of the reasons for that may lie in the attention module over the temporal tokens of the lagged time series, which we elaborate more with the datasets support in Section E.3.

It is also notable that applying FFN on both dimensions can also lead to fair performance on datasets with small variate numbers (such as Weather with 21 variates). Still, with the increasing of variate numbers in challenging multivariate forecasting tasks, the importance of capturing multivariate correlations is ever more highlighted. We note that the heterogeneity of variates can be hardly considered by the vanilla Transformer. During embedding, the variates are projected into indistinguishable channels, which ignores the inconsistent physical measurements and thus fails to maintain the independence of variates, let alone capture and utilize the multivariate correlation. Consequently, by incorporating the advanced attention module for the variate correlating, the first (iTransformer) and the fifth (attention on variates) designs perform more effectively in challenging multivariate datasets.

In a nutshell, both temporal dependencies and multivariate correlations are of importance for multivariate time series forecasting. The proposed iTransformer employing the self-attention module to disentangle the correlations between variate tokens proves to be more powerful and interpretable than feed-forward networks, thereby further boosting the performance on challenging multivariate datasets and enhancing the model capacity.

## C  HYPERPARAMETER SENSITIVITY

We evaluate the hyperparameter sensitivity of iTransformer with respect to the following factors: the learning rate $lr$, the number of Transformer blocks $L$, and the hidden dimension $D$ of variate tokens. The results are shown in Figure 9. We find that the learning rate, as the most common influencing factor, should be carefully selected when the number of variates is large (ECL, Traffic). The block number and hidden dimension are not essentially favored to be as large as possible in iTransformer.

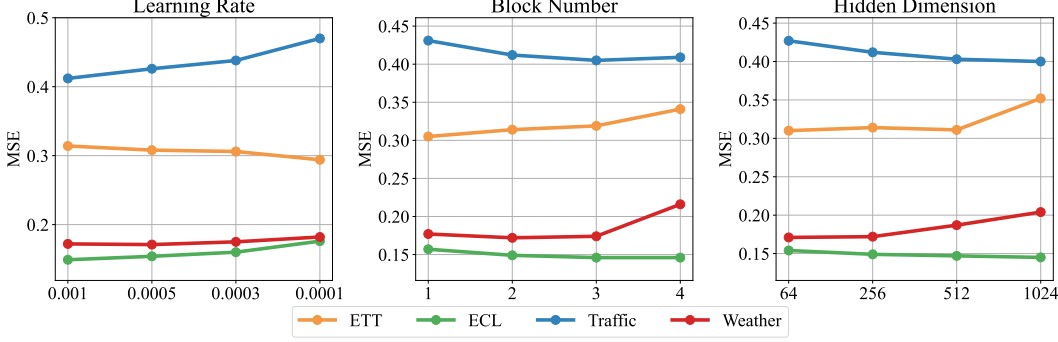

Figure 9: Hyperparameter sensitivity with respect to the learning rate, the number of Transformer blocks, and the hidden dimension of variate tokens. The results are recorded with the lookback window length $T = 96$ and the forecast window length $S = 96$.

## D  MODEL EFFICIENCY

We comprehensively compare the forecasting performance, training speed, and memory footprint of the following models: iTransformer, iTransformer with our efficient training strategy and iTransformer

Table 6: Full results of the ablation on iTransformer. We apply different components on the respective dimension to learn multivariate correlations (Variate) and series representations (Temporal), in addition to removing the specific component of Transformer.

| Design | Variate | Temporal | Prediction Lengths | ECL | | Traffic | | Weather | | Solar-Energy | |
|---|---|---|---|---|---|---|---|---|---|---|---|
| | | | | MSE | MAE | MSE | MAE | MSE | MAE | MSE | MAE |
| **iTransformer** | **Attention** | **FFN** | 96 | 0.148 | 0.240 | 0.395 | 0.268 | 0.174 | 0.214 | 0.203 | 0.237 |
| | | | 192 | 0.162 | 0.253 | 0.417 | 0.276 | 0.221 | 0.254 | 0.233 | 0.261 |
| | | | 336 | 0.178 | 0.269 | 0.433 | 0.283 | 0.278 | 0.296 | 0.248 | 0.273 |
| | | | 720 | 0.225 | 0.317 | 0.467 | 0.302 | 0.358 | 0.349 | 0.249 | 0.275 |
| | | | Avg | **0.178** | **0.270** | **0.428** | **0.282** | 0.258 | 0.279 | **0.233** | **0.262** |
| Replace | Attention | Attention | 96 | 0.161 | 0.263 | 1.021 | 0.581 | 0.168 | 0.213 | 0.227 | 0.270 |
| | | | 192 | 0.180 | 0.280 | 0.834 | 0.447 | 0.217 | 0.256 | 0.255 | 0.292 |
| | | | 336 | 0.194 | 0.296 | 0.906 | 0.493 | 0.277 | 0.299 | 0.279 | 0.301 |
| | | | 720 | 0.238 | 0.331 | 0.892 | 0.477 | 0.356 | 0.351 | 0.283 | 0.300 |
| | | | Avg | 0.193 | 0.293 | 0.913 | 0.500 | 0.255 | 0.280 | 0.261 | 0.291 |
| | FFN | Attention | 96 | 0.169 | 0.270 | 0.907 | 0.540 | 0.176 | 0.221 | 0.247 | 0.299 |
| | | | 192 | 0.189 | 0.292 | 0.839 | 0.489 | 0.224 | 0.261 | 0.275 | 0.305 |
| | | | 336 | 0.204 | 0.304 | 0.248 | 0.364 | 0.279 | 0.301 | 0.317 | 0.337 |
| | | | 720 | 0.245 | 0.335 | 1.059 | 0.606 | 0.354 | 0.347 | 0.301 | 0.329 |
| | | | Avg | 0.202 | 0.300 | 0.863 | 0.499 | 0.258 | 0.283 | 0.285 | 0.317 |
| | FFN | FFN | 96 | 0.159 | 0.261 | 0.606 | 0.342 | 0.162 | 0.207 | 0.237 | 0.277 |
| | | | 192 | 0.171 | 0.271 | 0.559 | 0.342 | 0.211 | 0.252 | 0.273 | 0.293 |
| | | | 336 | 0.187 | 0.287 | 0.569 | 0.348 | 0.270 | 0.293 | 0.284 | 0.287 |
| | | | 720 | 0.211 | 0.307 | 0.664 | 0.359 | 0.349 | 0.345 | 0.284 | 0.289 |
| | | | Avg | 0.182 | 0.287 | 0.599 | 0.348 | **0.248** | **0.274** | 0.269 | 0.287 |
| w/o | Attention | w/o | 96 | 0.163 | 0.254 | 0.427 | 0.296 | 0.177 | 0.219 | 0.226 | 0.266 |
| | | | 192 | 0.174 | 0.263 | 0.446 | 0.300 | 0.226 | 0.259 | 0.255 | 0.288 |
| | | | 336 | 0.191 | 0.280 | 0.459 | 0.306 | 0.281 | 0.298 | 0.275 | 0.301 |
| | | | 720 | 0.228 | 0.315 | 0.492 | 0.324 | 0.359 | 0.249 | 0.275 | 0.301 |
| | | | Avg | 0.189 | 0.278 | 0.456 | 0.306 | 0.261 | 0.281 | 0.258 | 0.289 |
| | w/o | FFN | 96 | 0.169 | 0.253 | 0.437 | 0.283 | 0.183 | 0.220 | 0.228 | 0.263 |
| | | | 192 | 0.177 | 0.261 | 0.449 | 0.287 | 0.231 | 0.262 | 0.261 | 0.283 |
| | | | 336 | 0.194 | 0.278 | 0.464 | 0.294 | 0.285 | 0.300 | 0.279 | 0.294 |
| | | | 720 | 0.233 | 0.311 | 0.496 | 0.313 | 0.362 | 0.350 | 0.276 | 0.291 |
| | | | Avg | 0.193 | 0.276 | 0.461 | 0.294 | 0.265 | 0.283 | 0.261 | 0.283 |

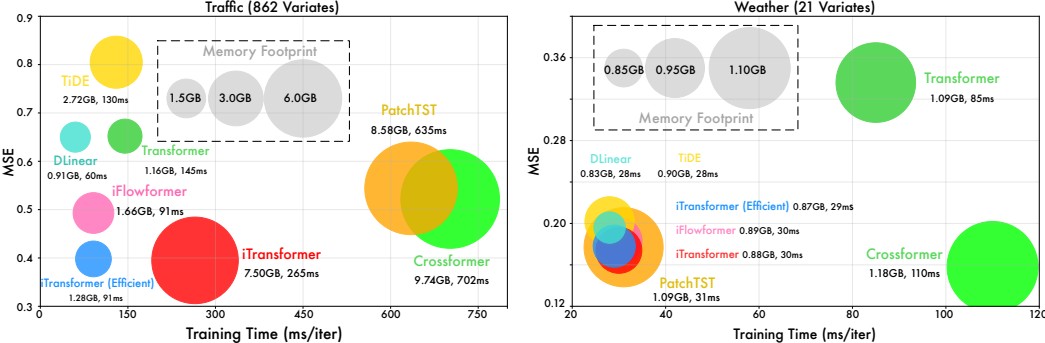

Figure 10: Model efficiency comparison under input-96-predict-96 of Weather and Traffic.

with the efficient flow attention module (Wu et al., 2022); linear models: DLinear (Zeng et al., 2023) and TiDE (Das et al., 2023); Transformers: Transformer (Vaswani et al., 2017), PatchTST (Nie et al., 2023), and Crossformer (Zhang & Yan, 2023). The results are recorded with the official model configuration and the same batch size. In Figure 10, we compare the efficiency under two representative datasets (21 variates in Weather and 862 in Traffic) with 96 time steps for lookback.

In a nutshell, the efficiency of iTransformer exceeds other Transformers in datasets with a relatively small number of variates (Weather). In datasets with numerous variates (Traffic), the memory footprints are basically the same as Transformers variates, but iTransformer can be trained faster. Based on the complexity of $\mathcal{O}(N^2)$ of the attention module, where $N$ is the number of tokens, Transformer surpasses iTransformer on efficiency in this case because of $N = 96$ for the temporal token and $N = 862$ for the variate token. Meanwhile, iTransformer achieves better performance on numerous variates, since the multivariate correlations can be explicitly utilized. By adopting a linear-complexity attention (Wu et al., 2022) or the proposed efficient training strategy as mentioned in Figure 8 (trained on 20% variates and forecast all variates), iTransformer can enjoy a comparable speed and memory footprint with linear models. Also, the two strategies can be adopted together.

## E SHOWCASES

### E.1 VISUALIZATION OF MULTIVARIATE CORRELATIONS

By using the attention mechanism on variate tokens, the resulting learned map becomes more interpretable. To present an intuitive understanding of the multivariate correlations, we provide three randomly chosen case visualizations of the time series from Solar-Energy in Figure 11. We provide the Pearson Correlation coefficients of each variate of the raw series by the following equation:

$$\rho_{xy} = \frac{\sum_i (x_i - \bar{x})(y_i - \bar{y})}{\sqrt{\sum_i (x_i - \bar{x})^2}\sqrt{\sum_i (y_i - \bar{y})^2}},$$

where $x_i, y_i \in \mathbb{R}$ run through all time points of the paired variates to be correlated. All the cases have distinct multivariate correlations in the lookback and forecast window because the dataset exhibits obvious seasonal changes in the daytime and night. On the second row of each case, we provide the learned pre-Softmax maps of the self-attention module in both the first and the last layers. As we observe in the shallow attention layer (left), we find that the learned map is similar to the correlations of the raw lookback series. As we go deeper into the layers (right), the learned map gradually becomes more similar to the correlations of the future series to be predicted. This demonstrates that the inverted operation allows for interpretable attention in correlating, and that encoding of the past and decoding for the future are conducted through series representations during layer stacking.

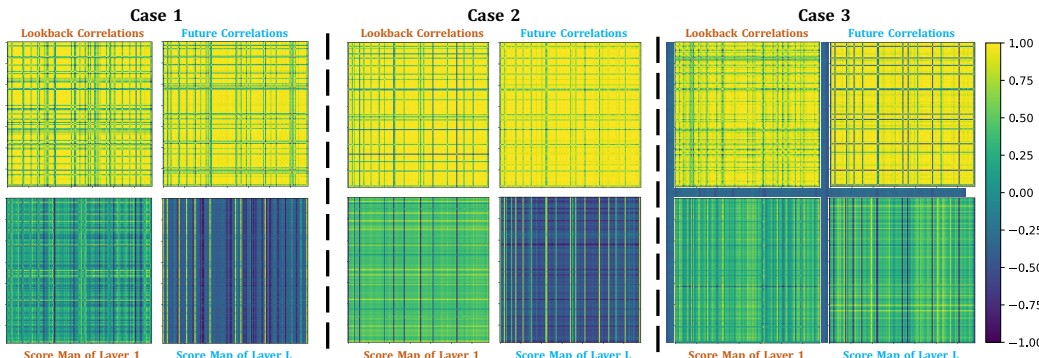

Figure 11: Multivariate correlations of the lookback series and future series and the learned score maps by inverted self-attention of different layers. Cases all come from the Solar-Energy dataset.

We present another interesting observation in Figure 12 to show that the attention module of iTransformer has enhanced interpretability. We provide randomly chosen multivariate time series from Market. In this dataset, each variate represents the monitored values of a service interface of a kind,

and the service can be further grouped into refined application categories. We divide these variates into corresponding applications (as listed on the top bar *App*), such that adjacent variates belong to the same application and we reveal the application index by the top bar.

We visualize the time series of the variates and plot the learned multivariate correlations with the marks of specific correlations between variates. On the one hand, we observe clear partitioning in the multivariate correlations map, indicating the grouping of variates. On the one hand, the marked correlation values can reflect the correlation of the raw series, where the similarity of variates from the same application becomes closer than the pairs from the different groups. Therefore, highly correlated variate will be leveraged for the next interaction and thus benefit for multivariate forecasting.

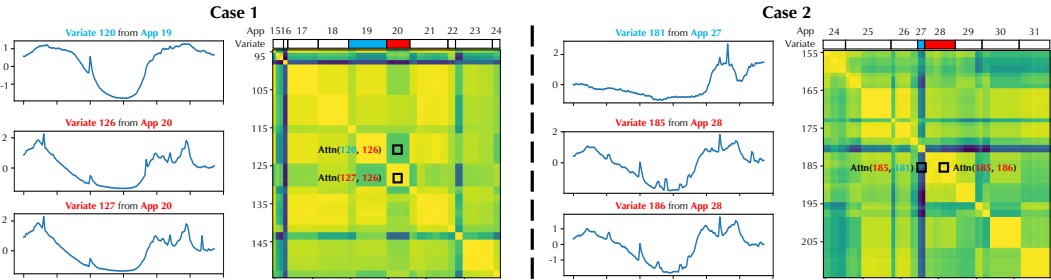

Figure 12: Visualization of the variates from the Market dataset and the learned multivariate correlations. Each variate represents the monitored interface values of an application, and the applications can be further grouped into refined categories. The color bar is shared with Figure 11.

## E.2 VISUALIZATION OF PREDICTION RESULTS

To provide a clear comparison among different models, we list supplementary prediction showcases of four representative datasets in Figures 13- 16, which are given by the following models: iTransfomrer, PatchTST (Nie et al., 2023), DLinear (Zeng et al., 2023), Crossformer (Zhang & Yan, 2023), Autoformer (Wu et al., 2021), Transformer (Vaswani et al., 2017). Among the various models, iTransformer predicts the most precise future series variations and exhibits superior performance.

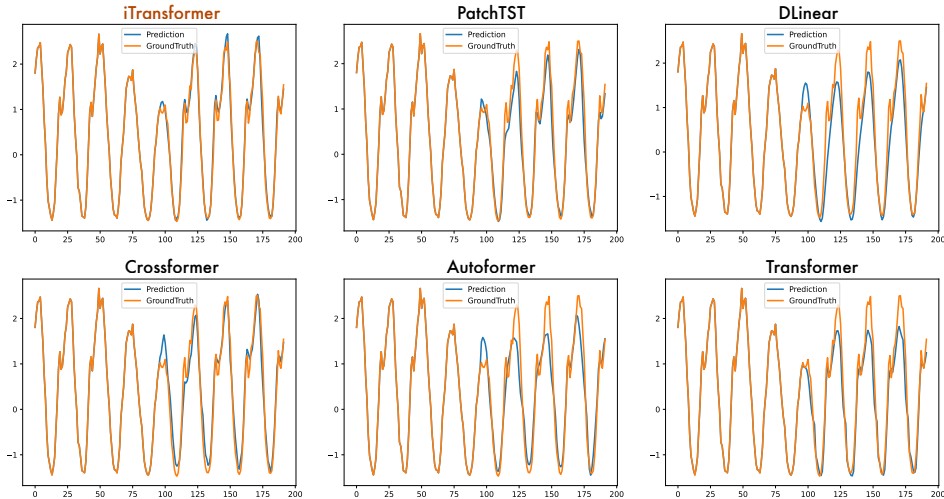

Figure 13: Visualization of input-96-predict-96 results on the Traffic dataset.

## E.3 RISKS OF EMBEDDING MULTIVARIATE POINTS OF A TIMESTAMP

As aforementioned, the embedding approach of the previous Transformer fuses multiple variates representing potentially delayed events and distinct physical measurements, which may fail to learn variate-centric representations and result in meaningless attention maps. We provide the visualization

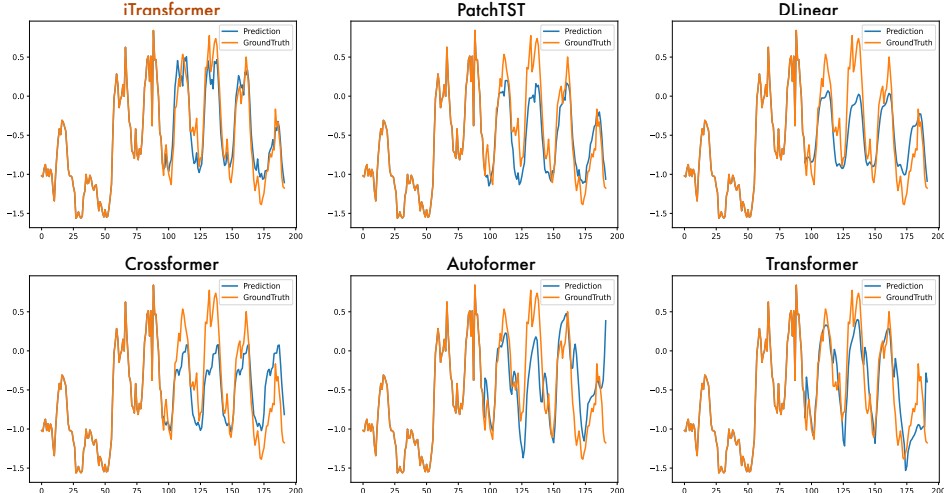

Figure 14: Visualization of input-96-predict-96 results on the ECL dataset.

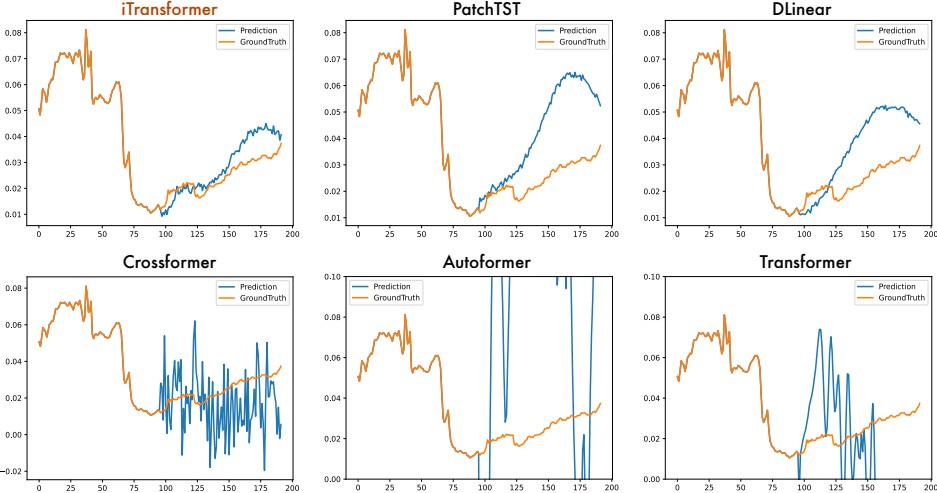

Figure 15: Visualization of input-96-predict-96 results on the Weather dataset.

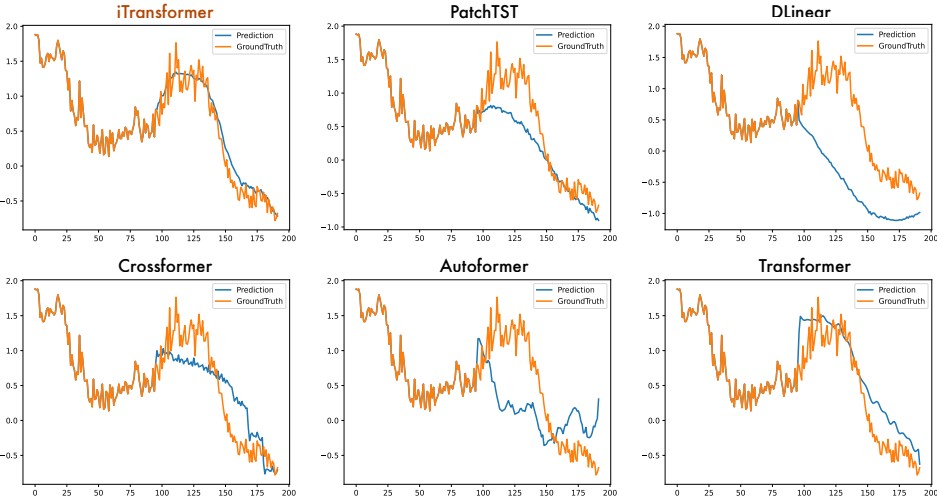

Figure 16: Visualization of input-96-predict-96 results on the PEMS dataset.

case of Traffic (Liu et al., 2022a), which is collected from sensors on Los Angeles city roads in different areas. As shown in Figure 17, we can observe a strong correlation between the multivariate time series of the dataset, while they also exhibit obvious phase offset, which is due to the systematical time lags in the road occupancy that each series describes. Since the sensors are installed in different areas of the highway, an event (such as a traffic jam) can affect road occupancy with different delays.

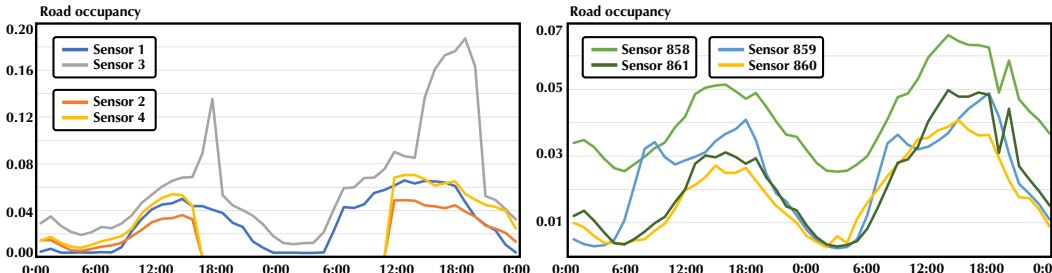

Figure 17: Visualization of partial variates of Traffic. We can observe that several series exhibit strong synchronization (such as Sensor 2 and Sensor 4), and there also exist obvious delays and advances between series (such as Sensor 1 and Sensor 2, Sensor 859 and Sensor 861).

Besides, we observe the significantly declined performance on the second and third designs of Traffic in Table 6, which apply attention to temporal tokens. In our opinion, capturing temporal dependencies by attention is not a big problem. But it is based on the fact that the time points of each timestamp essentially reflect the same event to enclose a semantic representation. Since there are inherent delays between the time points, the performance can degrade a lot because of the meaningless attention map, unless the model has an enlarged respective field to learn about the decay or causal process.

Other risks can be aroused from the distinct variate measurements, such as organizing together different meteorological indicators (the temperature and rainfall) in the Weather dataset (Wu et al., 2021), and the quantity and proportion of the same observation in ILI (Wu et al., 2023). Given these potential risks, iTransformer proposes a new paradigm that embeds the whole series as the variate token, which can be more robust to extensive real-world scenarios, such as delayed events, inconsistent measurements, irregular (unevenly spaced) time series, systematical delay of monitors, and the time interval of generating and recording different time series.

## F FULL RESULTS

### F.1 FULL PROMOTION RESULTS

We compare the performance of Transformer and iTransformer on all datasets in Table 7. Consistent and great promotions can be achieved, indicating that the attention and feed-forward network on the inverted dimensions greatly empower Transformers in multivariate time series forecasting, leaving an instructive direction to build up the foundation model of extensive time series data.

Table 7: Full performance comparison between the vanilla Transformer and the proposed iTransformer. The results are averaged from all four prediction lengths.

| Datasets | ETT | | ECL | | PEMS | | Solar-Energy | | Traffic | | Weather | |
|---|---|---|---|---|---|---|---|---|---|---|---|---|
| Metric | MSE | MAE | MSE | MAE | MSE | MAE | MSE | MAE | MSE | MAE | MSE | MAE |
| Transformer | 2.750 | 1.375 | 0.277 | 0.372 | 0.157 | 0.263 | 0.256 | 0.276 | 0.665 | 0.363 | 0.657 | 0.572 |
| **iTransformer** | **0.383** | **0.407** | **0.178** | **0.270** | **0.113** | **0.221** | **0.233** | **0.262** | **0.428** | **0.282** | **0.258** | **0.279** |
| Promotion | 86.1% | 70.4% | 35.6% | 27.4% | 28.0% | 16.0% | 9.0% | 5.1% | 35.6% | 22.3% | 60.2% | 50.8% |

### F.2 FULL FRAMEWORK GENERALITY RESULTS

We apply the proposed inverting framework to Transformer and its variants: Transformer (Vaswani et al., 2017), Reformer (Kitaev et al., 2020), Informer (Li et al., 2021), Flowformer (Wu et al.,

2022), Flashformer (Dao et al., 2022). The averaged results are shown in Table 2 due to the limited pages. We provide the supplementary forecasting results in Table 8. The results demonstrate that our iTransformers framework can consistently promote these Transformer variants, and take advantage of the booming efficient attention mechanisms.

Table 8: Full results of Transformers with our inverted framework. Flashformer means Transformer equipped with the hardware-accelerated FlashAttention (Dao et al., 2022).

| Models | | | Transformer (2017) | | Reformer (2020) | | Informer (2021) | | Flowformer (2022) | | Flashformer (2022) | |
|---|---|---|---|---|---|---|---|---|---|---|---|---|
| Metric | | | MSE | MAE | MSE | MAE | MSE | MAE | MSE | MAE | MSE | MAE |
| ECL | Original | 96 | 0.260 | 0.358 | 0.312 | 0.402 | 0.274 | 0.368 | 0.215 | 0.320 | 0.259 | 0.357 |
| | | 192 | 0.266 | 0.367 | 0.348 | 0.433 | 0.296 | 0.386 | 0.259 | 0.355 | 0.274 | 0.374 |
| | | 336 | 0.280 | 0.375 | 0.350 | 0.433 | 0.300 | 0.394 | 0.296 | 0.383 | 0.310 | 0.396 |
| | | 720 | 0.302 | 0.386 | 0.340 | 0.420 | 0.373 | 0.439 | 0.296 | 0.380 | 0.298 | 0.383 |
| | | Avg | 0.277 | 0.372 | 0.338 | 0.422 | 0.311 | 0.397 | 0.267 | 0.359 | 0.285 | 0.377 |
| | +Inverted | 96 | 0.148 | 0.240 | 0.182 | 0.275 | 0.190 | 0.286 | 0.183 | 0.267 | 0.178 | 0.265 |
| | | 192 | 0.162 | 0.253 | 0.192 | 0.286 | 0.201 | 0.297 | 0.192 | 0.277 | 0.189 | 0.276 |
| | | 336 | 0.178 | 0.269 | 0.210 | 0.304 | 0.218 | 0.315 | 0.210 | 0.295 | 0.207 | 0.294 |
| | | 720 | 0.225 | 0.317 | 0.249 | 0.339 | 0.255 | 0.347 | 0.255 | 0.332 | 0.251 | 0.329 |
| | | Avg | **0.178** | **0.270** | **0.208** | **0.301** | **0.216** | **0.311** | **0.210** | **0.293** | **0.206** | **0.291** |
| Traffic | Original | 96 | 0.647 | 0.357 | 0.732 | 0.423 | 0.719 | 0.391 | 0.691 | 0.393 | 0.641 | 0.348 |
| | | 192 | 0.649 | 0.356 | 0.733 | 0.420 | 0.696 | 0.379 | 0.729 | 0.419 | 0.648 | 0.358 |
| | | 336 | 0.667 | 0.364 | 0.742 | 0.420 | 0.777 | 0.420 | 0.756 | 0.423 | 0.670 | 0.364 |
| | | 720 | 0.697 | 0.376 | 0.755 | 0.432 | 0.864 | 0.472 | 0.825 | 0.449 | 0.673 | 0.354 |
| | | Avg | 0.665 | 0.363 | 0.741 | 0.422 | 0.764 | 0.416 | 0.750 | 0.421 | 0.658 | 0.356 |
| | +Inverted | 96 | 0.395 | 0.268 | 0.617 | 0.356 | 0.632 | 0.367 | 0.493 | 0.339 | 0.464 | 0.320 |
| | | 192 | 0.417 | 0.276 | 0.629 | 0.361 | 0.641 | 0.370 | 0.506 | 0.345 | 0.479 | 0.326 |
| | | 336 | 0.433 | 0.283 | 0.648 | 0.370 | 0.663 | 0.379 | 0.526 | 0.355 | 0.501 | 0.337 |
| | | 720 | 0.467 | 0.302 | 0.694 | 0.394 | 0.713 | 0.405 | 0.572 | 0.381 | 0.524 | 0.350 |
| | | Avg | **0.428** | **0.282** | **0.647** | **0.370** | **0.662** | **0.380** | **0.524** | **0.355** | **0.492** | **0.333** |
| Weather | Original | 96 | 0.395 | 0.427 | 0.689 | 0.596 | 0.300 | 0.384 | 0.182 | 0.233 | 0.388 | 0.425 |
| | | 192 | 0.619 | 0.560 | 0.752 | 0.638 | 0.598 | 0.544 | 0.250 | 0.288 | 0.619 | 0.560 |
| | | 336 | 0.689 | 0.594 | 0.639 | 0.596 | 0.578 | 0.523 | 0.309 | 0.329 | 0.698 | 0.600 |
| | | 720 | 0.926 | 0.710 | 1.130 | 0.792 | 1.059 | 0.741 | 0.404 | 0.385 | 0.930 | 0.711 |
| | | Avg | 0.657 | 0.572 | 0.803 | 0.656 | 0.634 | 0.548 | 0.286 | 0.308 | 0.659 | 0.574 |
| | +Inverted | 96 | 0.174 | 0.214 | 0.169 | 0.225 | 0.180 | 0.251 | 0.183 | 0.223 | 0.177 | 0.218 |
| | | 192 | 0.221 | 0.254 | 0.213 | 0.265 | 0.244 | 0.318 | 0.231 | 0.262 | 0.229 | 0.261 |
| | | 336 | 0.278 | 0.296 | 0.268 | 0.317 | 0.282 | 0.343 | 0.286 | 0.301 | 0.283 | 0.300 |
| | | 720 | 0.358 | 0.349 | 0.340 | 0.361 | 0.377 | 0.409 | 0.363 | 0.352 | 0.359 | 0.251 |
| | | Avg | **0.258** | **0.279** | **0.248** | **0.292** | **0.271** | **0.330** | **0.266** | **0.285** | **0.262** | **0.282** |

## F.3 FULL RESULTS OF VARIATE GENERALIZATION

We divide the variates of each dataset into five folders, train models with only 20% of variates of one folder, and directly forecast all variates without fine-tuning. We adopt two strategies for Transformers to generalize on unseen variates: (1) **CI-Transformers** (Nie et al., 2023): Channel Independence regards each variate of time series as independent channels, and trains with a shared backbone. During inference, the model predicts variates one by one, but the procedure can be time-consuming. (2) **iTransformers**: with the flexibility of the attention mechanism that the number of input tokens can be dynamically changeable, the amount of variates as tokens is no longer restricted and thus feasible to vary from training and inference, and can even allow the model to be trained on arbitrary variates.

As shown in Table 18, iTransformers can be naturally trained with 20% variates and accomplish forecast on all variates with the ability to learn transferable representations.

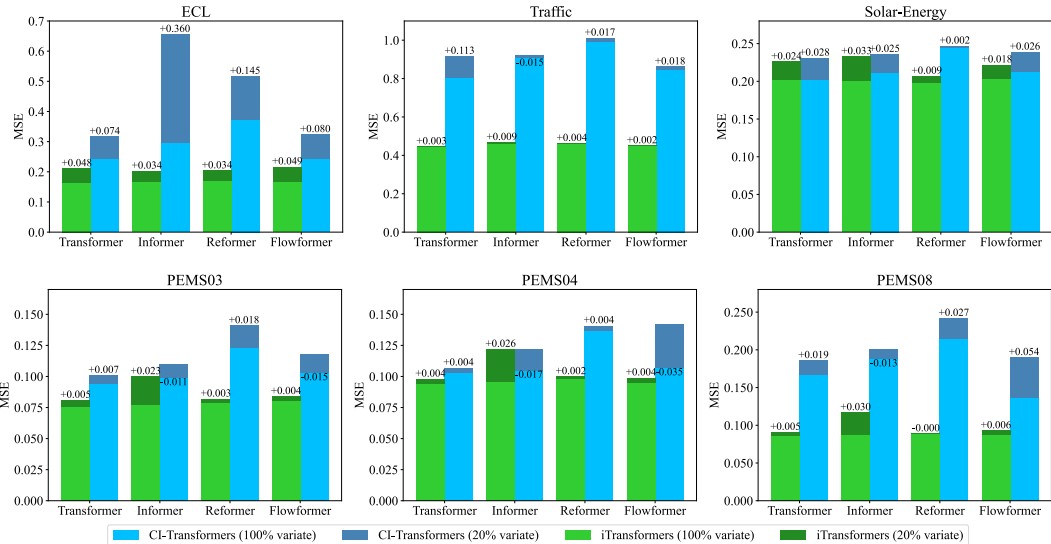

Figure 18: Full performance of generalization on unseen variates, comparing the iTransformers with CI-Transfomers. We divide the variates of each dataset into five folders, train with 20% variates, and use the trained model to forecast all varieties. We plot the averaged results of all five folders.

## F.4 FULL FORECASTING RESULTS

The full multivariate forecasting results are provided in the following section due to the space limitation of the main text. We extensively evaluate competitive counterparts on challenging forecasting tasks. Table 9 contains the forecasting results on the four public subsets from PEMS (Liu et al., 2022a). Table 10 contains the detailed results of all prediction lengths of the nine well-acknowledged forecasting benchmarks. And Table 11 records the Market results for Alipay server load forecasting. The proposed model achieves comprehensive state-of-the-art in real-world forecasting applications.

Table 9: Full results of the PEMS forecasting task. We compare extensive competitive models under different prediction lengths following the setting of SCINet (2022a). The input length is set to 96 for all baselines. *Avg* means the average results from all four prediction lengths.

| Models | | iTransformer (Ours) | | RLinear (2023) | | PatchTST (2023) | | Crossformer (2023) | | TiDE (2023) | | TimesNet (2023) | | DLinear (2023) | | SCINet (2022a) | | FEDformer (2022) | | Stationary (2022b) | | Autoformer (2021) | |
|---|---|---|---|---|---|---|---|---|---|---|---|---|---|---|---|---|---|---|---|---|---|---|---|
| Metric | | MSE | MAE | MSE | MAE | MSE | MAE | MSE | MAE | MSE | MAE | MSE | MAE | MSE | MAE | MSE | MAE | MSE | MAE | MSE | MAE | MSE | MAE |
| PEMS03 | 12 | 0.071 | 0.174 | 0.126 | 0.236 | 0.099 | 0.216 | 0.090 | 0.203 | 0.178 | 0.305 | 0.085 | 0.192 | 0.122 | 0.243 | 0.066 | 0.172 | 0.126 | 0.251 | 0.081 | 0.188 | 0.272 | 0.385 |
| | 24 | 0.093 | 0.201 | 0.246 | 0.334 | 0.142 | 0.259 | 0.121 | 0.240 | 0.257 | 0.371 | 0.118 | 0.223 | 0.201 | 0.317 | 0.085 | 0.198 | 0.149 | 0.275 | 0.105 | 0.214 | 0.334 | 0.440 |
| | 48 | 0.125 | 0.236 | 0.551 | 0.529 | 0.211 | 0.319 | 0.202 | 0.317 | 0.379 | 0.463 | 0.155 | 0.260 | 0.333 | 0.425 | 0.127 | 0.238 | 0.227 | 0.348 | 0.154 | 0.257 | 1.032 | 0.782 |
| | 96 | 0.164 | 0.275 | 1.057 | 0.787 | 0.269 | 0.370 | 0.262 | 0.367 | 0.490 | 0.539 | 0.228 | 0.317 | 0.457 | 0.515 | 0.178 | 0.287 | 0.348 | 0.434 | 0.247 | 0.336 | 1.031 | 0.796 |
| | Avg | 0.113 | 0.221 | 0.495 | 0.472 | 0.180 | 0.291 | 0.169 | 0.281 | 0.326 | 0.419 | 0.147 | 0.248 | 0.278 | 0.375 | 0.114 | 0.224 | 0.213 | 0.327 | 0.147 | 0.249 | 0.667 | 0.601 |
| PEMS04 | 12 | 0.078 | 0.183 | 0.138 | 0.252 | 0.105 | 0.224 | 0.098 | 0.218 | 0.219 | 0.340 | 0.087 | 0.195 | 0.148 | 0.272 | 0.073 | 0.177 | 0.138 | 0.262 | 0.088 | 0.196 | 0.424 | 0.491 |
| | 24 | 0.095 | 0.205 | 0.258 | 0.348 | 0.153 | 0.275 | 0.131 | 0.256 | 0.292 | 0.398 | 0.103 | 0.215 | 0.224 | 0.340 | 0.084 | 0.193 | 0.177 | 0.293 | 0.104 | 0.216 | 0.459 | 0.509 |
| | 48 | 0.120 | 0.233 | 0.572 | 0.544 | 0.229 | 0.339 | 0.205 | 0.326 | 0.409 | 0.478 | 0.136 | 0.250 | 0.355 | 0.437 | 0.099 | 0.211 | 0.270 | 0.368 | 0.137 | 0.251 | 0.646 | 0.610 |
| | 96 | 0.150 | 0.262 | 1.137 | 0.820 | 0.291 | 0.389 | 0.402 | 0.457 | 0.492 | 0.532 | 0.190 | 0.303 | 0.452 | 0.504 | 0.114 | 0.227 | 0.341 | 0.427 | 0.186 | 0.297 | 0.912 | 0.748 |
| | Avg | 0.111 | 0.221 | 0.526 | 0.491 | 0.195 | 0.307 | 0.209 | 0.314 | 0.353 | 0.437 | 0.129 | 0.241 | 0.295 | 0.388 | 0.092 | 0.202 | 0.231 | 0.337 | 0.127 | 0.240 | 0.610 | 0.590 |
| PEMS07 | 12 | 0.067 | 0.165 | 0.118 | 0.235 | 0.095 | 0.207 | 0.094 | 0.200 | 0.173 | 0.304 | 0.082 | 0.181 | 0.115 | 0.242 | 0.068 | 0.171 | 0.109 | 0.225 | 0.083 | 0.185 | 0.199 | 0.336 |
| | 24 | 0.088 | 0.190 | 0.242 | 0.341 | 0.150 | 0.262 | 0.139 | 0.247 | 0.271 | 0.383 | 0.101 | 0.204 | 0.210 | 0.329 | 0.119 | 0.225 | 0.125 | 0.244 | 0.102 | 0.207 | 0.323 | 0.420 |
| | 48 | 0.110 | 0.215 | 0.562 | 0.541 | 0.253 | 0.340 | 0.311 | 0.369 | 0.446 | 0.495 | 0.134 | 0.238 | 0.398 | 0.458 | 0.149 | 0.237 | 0.165 | 0.288 | 0.136 | 0.240 | 0.390 | 0.470 |
| | 96 | 0.139 | 0.245 | 1.096 | 0.795 | 0.346 | 0.404 | 0.396 | 0.442 | 0.628 | 0.577 | 0.181 | 0.279 | 0.594 | 0.553 | 0.141 | 0.234 | 0.262 | 0.376 | 0.187 | 0.287 | 0.554 | 0.578 |
| | Avg | 0.101 | 0.204 | 0.504 | 0.478 | 0.211 | 0.303 | 0.235 | 0.315 | 0.380 | 0.440 | 0.124 | 0.225 | 0.329 | 0.395 | 0.119 | 0.234 | 0.165 | 0.283 | 0.127 | 0.230 | 0.367 | 0.451 |
| PEMS08 | 12 | 0.079 | 0.182 | 0.133 | 0.247 | 0.168 | 0.232 | 0.165 | 0.214 | 0.227 | 0.343 | 0.112 | 0.212 | 0.154 | 0.276 | 0.087 | 0.184 | 0.173 | 0.273 | 0.109 | 0.207 | 0.436 | 0.485 |
| | 24 | 0.115 | 0.219 | 0.249 | 0.343 | 0.224 | 0.281 | 0.215 | 0.260 | 0.318 | 0.409 | 0.141 | 0.238 | 0.248 | 0.353 | 0.122 | 0.221 | 0.210 | 0.301 | 0.140 | 0.236 | 0.467 | 0.502 |
| | 48 | 0.186 | 0.235 | 0.569 | 0.544 | 0.321 | 0.354 | 0.315 | 0.355 | 0.497 | 0.510 | 0.198 | 0.283 | 0.440 | 0.470 | 0.189 | 0.270 | 0.320 | 0.394 | 0.211 | 0.294 | 0.966 | 0.733 |
| | 96 | 0.221 | 0.267 | 1.166 | 0.814 | 0.408 | 0.417 | 0.377 | 0.397 | 0.721 | 0.592 | 0.320 | 0.351 | 0.674 | 0.565 | 0.236 | 0.300 | 0.442 | 0.465 | 0.345 | 0.367 | 1.385 | 0.915 |
| | Avg | 0.150 | 0.226 | 0.529 | 0.487 | 0.280 | 0.321 | 0.268 | 0.307 | 0.441 | 0.464 | 0.193 | 0.271 | 0.379 | 0.416 | 0.158 | 0.244 | 0.286 | 0.358 | 0.201 | 0.276 | 0.814 | 0.659 |
| 1st Count | | 13 | 13 | 0 | 0 | 0 | 0 | 0 | 0 | 0 | 0 | 0 | 0 | 0 | 0 | 7 | 7 | 0 | 0 | 0 | 0 | 0 | 0 |

Table 10: Full results of the long-term forecasting task. We compare extensive competitive models under different prediction lengths following the setting of TimesNet (2023). The input sequence length is set to 96 for all baselines. *Avg* means the average results from all four prediction lengths.

| Models | | iTransformer (Ours) | | RLinear (2023) | | PatchTST (2023) | | Crossformer (2023) | | TiDE (2023) | | TimesNet (2023) | | DLinear (2023) | | SCINet (2022a) | | FEDformer (2022) | | Stationary (2022b) | | Autoformer (2021) | |
|---|---|---|---|---|---|---|---|---|---|---|---|---|---|---|---|---|---|---|---|---|---|---|---|
| Metric | | MSE | MAE | MSE | MAE | MSE | MAE | MSE | MAE | MSE | MAE | MSE | MAE | MSE | MAE | MSE | MAE | MSE | MAE | MSE | MAE | MSE | MAE |
| ETTm1 | 96 | 0.334 | 0.368 | 0.355 | 0.376 | 0.329 | 0.367 | 0.404 | 0.426 | 0.364 | 0.387 | 0.338 | 0.375 | 0.345 | 0.372 | 0.418 | 0.438 | 0.379 | 0.419 | 0.386 | 0.398 | 0.505 | 0.475 |
| | 192 | 0.377 | 0.391 | 0.391 | 0.392 | 0.367 | 0.385 | 0.450 | 0.451 | 0.398 | 0.404 | 0.374 | 0.387 | 0.380 | 0.389 | 0.439 | 0.450 | 0.426 | 0.441 | 0.459 | 0.444 | 0.553 | 0.496 |
| | 336 | 0.426 | 0.420 | 0.424 | 0.415 | 0.399 | 0.410 | 0.532 | 0.515 | 0.428 | 0.425 | 0.410 | 0.411 | 0.413 | 0.413 | 0.490 | 0.485 | 0.445 | 0.459 | 0.495 | 0.464 | 0.621 | 0.537 |
| | 720 | 0.491 | 0.459 | 0.487 | 0.450 | 0.454 | 0.439 | 0.666 | 0.589 | 0.487 | 0.461 | 0.478 | 0.450 | 0.474 | 0.453 | 0.595 | 0.550 | 0.543 | 0.490 | 0.585 | 0.516 | 0.671 | 0.561 |
| | Avg | 0.407 | 0.410 | 0.414 | 0.407 | 0.387 | 0.400 | 0.513 | 0.496 | 0.419 | 0.419 | 0.400 | 0.406 | 0.403 | 0.407 | 0.485 | 0.481 | 0.448 | 0.452 | 0.481 | 0.456 | 0.588 | 0.517 |
| ETTm2 | 96 | 0.180 | 0.264 | 0.182 | 0.265 | 0.175 | 0.259 | 0.287 | 0.366 | 0.207 | 0.305 | 0.187 | 0.267 | 0.193 | 0.292 | 0.286 | 0.377 | 0.203 | 0.287 | 0.192 | 0.274 | 0.255 | 0.339 |
| | 192 | 0.250 | 0.309 | 0.246 | 0.304 | 0.241 | 0.302 | 0.414 | 0.492 | 0.290 | 0.364 | 0.249 | 0.309 | 0.284 | 0.362 | 0.399 | 0.445 | 0.269 | 0.328 | 0.280 | 0.339 | 0.281 | 0.340 |
| | 336 | 0.311 | 0.348 | 0.307 | 0.342 | 0.305 | 0.343 | 0.597 | 0.542 | 0.377 | 0.422 | 0.321 | 0.351 | 0.369 | 0.427 | 0.637 | 0.591 | 0.325 | 0.366 | 0.334 | 0.361 | 0.339 | 0.372 |
| | 720 | 0.412 | 0.407 | 0.407 | 0.398 | 0.402 | 0.400 | 1.730 | 1.042 | 0.558 | 0.524 | 0.408 | 0.403 | 0.554 | 0.522 | 0.960 | 0.735 | 0.421 | 0.415 | 0.417 | 0.413 | 0.433 | 0.432 |
| | Avg | 0.288 | 0.332 | 0.286 | 0.327 | 0.281 | 0.326 | 0.757 | 0.610 | 0.358 | 0.404 | 0.291 | 0.333 | 0.350 | 0.401 | 0.571 | 0.537 | 0.305 | 0.349 | 0.306 | 0.347 | 0.327 | 0.371 |
| ETTh1 | 96 | 0.386 | 0.405 | 0.386 | 0.395 | 0.414 | 0.419 | 0.423 | 0.448 | 0.479 | 0.464 | 0.384 | 0.402 | 0.386 | 0.400 | 0.654 | 0.599 | 0.376 | 0.419 | 0.513 | 0.491 | 0.449 | 0.459 |
| | 192 | 0.441 | 0.436 | 0.437 | 0.424 | 0.460 | 0.445 | 0.471 | 0.474 | 0.525 | 0.492 | 0.436 | 0.429 | 0.437 | 0.432 | 0.719 | 0.631 | 0.420 | 0.448 | 0.534 | 0.504 | 0.500 | 0.482 |
| | 336 | 0.487 | 0.458 | 0.479 | 0.446 | 0.501 | 0.466 | 0.570 | 0.546 | 0.565 | 0.515 | 0.491 | 0.469 | 0.481 | 0.459 | 0.778 | 0.659 | 0.459 | 0.465 | 0.588 | 0.535 | 0.521 | 0.496 |
| | 720 | 0.503 | 0.491 | 0.481 | 0.470 | 0.500 | 0.488 | 0.653 | 0.621 | 0.594 | 0.558 | 0.521 | 0.500 | 0.519 | 0.516 | 0.836 | 0.699 | 0.506 | 0.507 | 0.643 | 0.616 | 0.514 | 0.512 |
| | Avg | 0.454 | 0.447 | 0.446 | 0.434 | 0.469 | 0.454 | 0.529 | 0.522 | 0.541 | 0.507 | 0.458 | 0.450 | 0.456 | 0.452 | 0.747 | 0.647 | 0.440 | 0.460 | 0.570 | 0.537 | 0.496 | 0.487 |
| ETTh2 | 96 | 0.297 | 0.349 | 0.288 | 0.338 | 0.302 | 0.348 | 0.745 | 0.584 | 0.400 | 0.440 | 0.340 | 0.374 | 0.333 | 0.387 | 0.707 | 0.621 | 0.358 | 0.397 | 0.476 | 0.458 | 0.346 | 0.388 |
| | 192 | 0.380 | 0.400 | 0.374 | 0.390 | 0.388 | 0.400 | 0.877 | 0.656 | 0.528 | 0.509 | 0.402 | 0.414 | 0.477 | 0.476 | 0.860 | 0.689 | 0.429 | 0.439 | 0.512 | 0.493 | 0.456 | 0.452 |
| | 336 | 0.428 | 0.432 | 0.415 | 0.426 | 0.426 | 0.433 | 1.043 | 0.731 | 0.643 | 0.571 | 0.452 | 0.452 | 0.594 | 0.541 | 1.000 | 0.744 | 0.496 | 0.487 | 0.552 | 0.551 | 0.482 | 0.486 |
| | 720 | 0.427 | 0.445 | 0.420 | 0.440 | 0.431 | 0.446 | 1.104 | 0.763 | 0.874 | 0.679 | 0.462 | 0.468 | 0.831 | 0.657 | 1.249 | 0.838 | 0.463 | 0.474 | 0.562 | 0.560 | 0.515 | 0.511 |
| | Avg | 0.383 | 0.407 | 0.374 | 0.398 | 0.387 | 0.407 | 0.942 | 0.684 | 0.611 | 0.550 | 0.414 | 0.427 | 0.559 | 0.515 | 0.954 | 0.723 | 0.437 | 0.449 | 0.526 | 0.516 | 0.450 | 0.459 |
| ECL | 96 | 0.148 | 0.240 | 0.201 | 0.281 | 0.181 | 0.270 | 0.219 | 0.314 | 0.237 | 0.329 | 0.168 | 0.272 | 0.197 | 0.282 | 0.247 | 0.345 | 0.193 | 0.308 | 0.169 | 0.273 | 0.201 | 0.317 |
| | 192 | 0.162 | 0.253 | 0.201 | 0.283 | 0.188 | 0.274 | 0.231 | 0.322 | 0.236 | 0.330 | 0.184 | 0.289 | 0.196 | 0.285 | 0.257 | 0.355 | 0.201 | 0.315 | 0.182 | 0.286 | 0.222 | 0.334 |
| | 336 | 0.178 | 0.269 | 0.215 | 0.298 | 0.204 | 0.293 | 0.246 | 0.337 | 0.249 | 0.344 | 0.198 | 0.300 | 0.209 | 0.301 | 0.269 | 0.369 | 0.214 | 0.329 | 0.200 | 0.304 | 0.231 | 0.338 |
| | 720 | 0.225 | 0.317 | 0.257 | 0.331 | 0.246 | 0.324 | 0.280 | 0.363 | 0.284 | 0.373 | 0.220 | 0.320 | 0.245 | 0.333 | 0.299 | 0.390 | 0.246 | 0.355 | 0.222 | 0.321 | 0.254 | 0.361 |
| | Avg | 0.178 | 0.270 | 0.219 | 0.298 | 0.205 | 0.290 | 0.244 | 0.334 | 0.251 | 0.344 | 0.192 | 0.295 | 0.212 | 0.300 | 0.268 | 0.365 | 0.214 | 0.327 | 0.193 | 0.296 | 0.227 | 0.338 |
| Exchange | 96 | 0.086 | 0.206 | 0.093 | 0.217 | 0.088 | 0.205 | 0.256 | 0.367 | 0.094 | 0.218 | 0.107 | 0.234 | 0.088 | 0.218 | 0.267 | 0.396 | 0.148 | 0.278 | 0.111 | 0.237 | 0.197 | 0.323 |
| | 192 | 0.177 | 0.299 | 0.184 | 0.307 | 0.176 | 0.299 | 0.470 | 0.509 | 0.184 | 0.307 | 0.226 | 0.344 | 0.176 | 0.315 | 0.351 | 0.459 | 0.271 | 0.315 | 0.219 | 0.335 | 0.300 | 0.369 |
| | 336 | 0.331 | 0.417 | 0.351 | 0.432 | 0.301 | 0.397 | 1.268 | 0.883 | 0.349 | 0.431 | 0.367 | 0.448 | 0.313 | 0.427 | 1.324 | 0.853 | 0.460 | 0.427 | 0.421 | 0.476 | 0.509 | 0.524 |
| | 720 | 0.847 | 0.691 | 0.886 | 0.714 | 0.901 | 0.714 | 1.767 | 1.068 | 0.852 | 0.698 | 0.964 | 0.746 | 0.839 | 0.695 | 1.058 | 0.797 | 1.195 | 0.695 | 1.092 | 0.769 | 1.447 | 0.941 |
| | Avg | 0.360 | 0.403 | 0.378 | 0.417 | 0.367 | 0.404 | 0.940 | 0.707 | 0.370 | 0.413 | 0.416 | 0.443 | 0.354 | 0.414 | 0.750 | 0.626 | 0.519 | 0.429 | 0.461 | 0.454 | 0.613 | 0.539 |
| Traffic | 96 | 0.395 | 0.268 | 0.649 | 0.389 | 0.462 | 0.295 | 0.522 | 0.290 | 0.805 | 0.493 | 0.593 | 0.321 | 0.650 | 0.396 | 0.788 | 0.499 | 0.587 | 0.366 | 0.612 | 0.338 | 0.613 | 0.388 |
| | 192 | 0.417 | 0.276 | 0.601 | 0.366 | 0.466 | 0.296 | 0.530 | 0.293 | 0.756 | 0.474 | 0.617 | 0.336 | 0.598 | 0.370 | 0.789 | 0.505 | 0.604 | 0.373 | 0.613 | 0.340 | 0.616 | 0.382 |
| | 336 | 0.433 | 0.283 | 0.609 | 0.369 | 0.482 | 0.304 | 0.558 | 0.305 | 0.762 | 0.477 | 0.629 | 0.336 | 0.605 | 0.373 | 0.797 | 0.508 | 0.621 | 0.383 | 0.618 | 0.328 | 0.622 | 0.337 |
| | 720 | 0.467 | 0.302 | 0.647 | 0.387 | 0.514 | 0.322 | 0.589 | 0.328 | 0.719 | 0.449 | 0.640 | 0.350 | 0.645 | 0.394 | 0.841 | 0.523 | 0.626 | 0.382 | 0.653 | 0.355 | 0.660 | 0.408 |
| | Avg | 0.428 | 0.282 | 0.626 | 0.378 | 0.481 | 0.304 | 0.550 | 0.304 | 0.760 | 0.473 | 0.620 | 0.336 | 0.625 | 0.383 | 0.804 | 0.509 | 0.610 | 0.376 | 0.624 | 0.340 | 0.628 | 0.379 |
| Weather | 96 | 0.174 | 0.214 | 0.192 | 0.232 | 0.177 | 0.218 | 0.158 | 0.230 | 0.202 | 0.261 | 0.172 | 0.220 | 0.196 | 0.255 | 0.221 | 0.306 | 0.217 | 0.296 | 0.173 | 0.223 | 0.266 | 0.336 |
| | 192 | 0.221 | 0.254 | 0.240 | 0.271 | 0.225 | 0.259 | 0.206 | 0.277 | 0.242 | 0.298 | 0.219 | 0.261 | 0.237 | 0.296 | 0.261 | 0.340 | 0.276 | 0.336 | 0.245 | 0.285 | 0.307 | 0.367 |
| | 336 | 0.278 | 0.296 | 0.292 | 0.307 | 0.278 | 0.297 | 0.272 | 0.335 | 0.287 | 0.335 | 0.280 | 0.306 | 0.283 | 0.335 | 0.309 | 0.378 | 0.339 | 0.380 | 0.321 | 0.338 | 0.359 | 0.395 |
| | 720 | 0.358 | 0.347 | 0.364 | 0.353 | 0.354 | 0.348 | 0.398 | 0.418 | 0.351 | 0.386 | 0.365 | 0.359 | 0.345 | 0.381 | 0.377 | 0.427 | 0.403 | 0.428 | 0.414 | 0.410 | 0.419 | 0.428 |
| | Avg | 0.258 | 0.278 | 0.272 | 0.291 | 0.259 | 0.281 | 0.259 | 0.315 | 0.271 | 0.320 | 0.259 | 0.287 | 0.265 | 0.317 | 0.292 | 0.363 | 0.309 | 0.360 | 0.288 | 0.314 | 0.338 | 0.382 |
| Solar-Energy | 96 | 0.203 | 0.237 | 0.322 | 0.339 | 0.234 | 0.286 | 0.310 | 0.331 | 0.312 | 0.399 | 0.250 | 0.292 | 0.290 | 0.378 | 0.237 | 0.344 | 0.242 | 0.342 | 0.215 | 0.249 | 0.884 | 0.711 |
| | 192 | 0.233 | 0.261 | 0.359 | 0.356 | 0.267 | 0.310 | 0.734 | 0.725 | 0.339 | 0.416 | 0.296 | 0.318 | 0.320 | 0.398 | 0.280 | 0.380 | 0.285 | 0.380 | 0.254 | 0.272 | 0.834 | 0.692 |
| | 336 | 0.248 | 0.273 | 0.397 | 0.369 | 0.290 | 0.315 | 0.750 | 0.735 | 0.368 | 0.430 | 0.319 | 0.330 | 0.353 | 0.415 | 0.304 | 0.389 | 0.282 | 0.376 | 0.290 | 0.296 | 0.941 | 0.723 |
| | 720 | 0.249 | 0.275 | 0.397 | 0.356 | 0.289 | 0.317 | 0.769 | 0.765 | 0.370 | 0.425 | 0.338 | 0.337 | 0.356 | 0.413 | 0.308 | 0.388 | 0.357 | 0.427 | 0.285 | 0.295 | 0.882 | 0.717 |
| | Avg | 0.233 | 0.262 | 0.369 | 0.356 | 0.270 | 0.307 | 0.641 | 0.639 | 0.347 | 0.417 | 0.301 | 0.319 | 0.330 | 0.401 | 0.282 | 0.375 | 0.291 | 0.381 | 0.261 | 0.381 | 0.885 | 0.711 |
| 1st Count | | 16 | 22 | 6 | 12 | 12 | 11 | 3 | 0 | 0 | 0 | 1 | 0 | 3 | 0 | 0 | 0 | 4 | 0 | 0 | 0 | 0 | 0 |

Table 11: Full results of the Market dataset. We compare extensive competitive models on the real-world transaction forecasting task. *Avg* means the average results from all prediction lengths.

| Models | | iTransformer (Ours) | | RLinear (2023) | | PatchTST (2023) | | Crossformer (2023) | | TiDE (2023) | | TimesNet (2023) | | DLinear (2023) | | SCINet (2022a) | | FEDformer (2022) | | Stationary (2022b) | | Autoformer (2021) | |
|---|---|---|---|---|---|---|---|---|---|---|---|---|---|---|---|---|---|---|---|---|---|---|---|
| Metric | | MSE | MAE | MSE | MAE | MSE | MAE | MSE | MAE | MSE | MAE | MSE | MAE | MSE | MAE | MSE | MAE | MSE | MAE | MSE | MAE | MSE | MAE |
| Merchant | 12 | **0.058** | **0.126** | 0.139 | 0.232 | 0.072 | 0.155 | 0.068 | 0.141 | 0.173 | 0.273 | 0.088 | 0.177 | 0.093 | 0.183 | 0.202 | 0.310 | 0.277 | 0.384 | 0.143 | 0.243 | 0.365 | 0.444 |
| | 24 | **0.066** | **0.138** | 0.155 | 0.250 | 0.079 | 0.164 | 0.091 | 0.161 | 0.170 | 0.274 | 0.103 | 0.195 | 0.105 | 0.200 | 0.215 | 0.323 | 0.268 | 0.378 | 0.167 | 0.270 | 0.669 | 0.636 |
| | 72 | **0.079** | **0.157** | 0.156 | 0.252 | 0.090 | 0.180 | 0.123 | 0.202 | 0.197 | 0.298 | 0.089 | 0.180 | 0.116 | 0.215 | 0.388 | 0.431 | 0.281 | 0.390 | 0.193 | 0.300 | 0.404 | 0.479 |
| | 144 | **0.086** | **0.167** | 0.157 | 0.253 | 0.093 | 0.185 | 0.185 | 0.218 | 0.208 | 0.311 | 0.091 | 0.183 | 0.124 | 0.225 | 0.459 | 0.477 | 0.359 | 0.453 | 0.183 | 0.294 | 0.536 | 0.566 |
| | Avg | **0.072** | **0.147** | 0.152 | 0.247 | 0.084 | 0.171 | 0.117 | 0.181 | 0.187 | 0.289 | 0.093 | 0.184 | 0.110 | 0.206 | 0.316 | 0.385 | 0.296 | 0.401 | 0.172 | 0.277 | 0.494 | 0.531 |
| Wealth | 12 | **0.189** | **0.205** | 0.479 | 0.411 | 0.255 | 0.250 | 0.270 | 0.208 | 0.486 | 0.427 | 0.275 | 0.277 | 0.380 | 0.355 | 0.525 | 0.451 | 0.553 | 0.508 | 0.355 | 0.332 | 0.653 | 0.555 |
| | 24 | **0.254** | **0.244** | 0.543 | 0.446 | 0.320 | 0.291 | 0.329 | 0.233 | 0.545 | 0.463 | 0.300 | 0.285 | 0.456 | 0.397 | 0.583 | 0.479 | 0.567 | 0.514 | 0.430 | 0.377 | 0.761 | 0.611 |
| | 72 | 0.421 | 0.327 | 0.634 | 0.481 | 0.459 | 0.360 | 0.484 | 0.324 | 0.651 | 0.510 | 0.384 | 0.326 | 0.555 | 0.438 | 0.761 | 0.558 | 0.636 | 0.548 | 0.573 | 0.454 | 0.857 | 0.658 |
| | 144 | 0.517 | 0.379 | 0.683 | 0.504 | 0.541 | 0.404 | 0.633 | 0.388 | 0.698 | 0.526 | 0.481 | 0.383 | 0.611 | 0.459 | 0.770 | 0.568 | 0.744 | 0.604 | 0.637 | 0.498 | 0.817 | 0.627 |
| | Avg | **0.345** | **0.289** | 0.585 | 0.461 | 0.394 | 0.326 | 0.429 | 0.288 | 0.595 | 0.481 | 0.360 | 0.318 | 0.501 | 0.412 | 0.660 | 0.514 | 0.625 | 0.543 | 0.499 | 0.415 | 0.772 | 0.612 |
| Finance | 12 | **0.123** | **0.170** | 0.329 | 0.304 | 0.164 | 0.206 | 4.630 | 0.520 | 0.512 | 0.350 | 0.465 | 0.291 | 0.321 | 0.271 | 1.865 | 0.602 | 1.537 | 0.538 | 0.537 | 0.384 | 1.651 | 0.593 |
| | 24 | **0.158** | **0.197** | 0.386 | 0.332 | 0.198 | 0.228 | 4.987 | 0.568 | 0.635 | 0.388 | 0.503 | 0.297 | 0.464 | 0.318 | 2.228 | 0.664 | 1.553 | 0.547 | 0.551 | 0.386 | 1.671 | 0.594 |
| | 72 | **0.212** | **0.240** | 0.436 | 0.353 | 0.268 | 0.273 | 5.631 | 0.675 | 1.239 | 0.490 | 0.534 | 0.310 | 0.986 | 0.423 | 3.084 | 0.793 | 1.612 | 0.554 | 2.004 | 0.853 | 2.054 | 0.758 |
| | 144 | **0.245** | **0.257** | 0.429 | 0.355 | 0.293 | 0.286 | 6.083 | 0.708 | 1.562 | 0.538 | 0.564 | 0.333 | 1.287 | 0.473 | 4.089 | 0.875 | 1.784 | 0.636 | 2.379 | 0.947 | 2.114 | 0.778 |
| | Avg | **0.184** | **0.216** | 0.395 | 0.336 | 0.231 | 0.248 | 5.333 | 0.618 | 0.987 | 0.442 | 0.516 | 0.308 | 0.765 | 0.372 | 2.817 | 0.734 | 1.621 | 0.569 | 1.368 | 0.643 | 1.872 | 0.681 |
| Terminal | 12 | **0.051** | **0.127** | 0.168 | 0.272 | 0.068 | 0.164 | 0.055 | 0.140 | 0.212 | 0.304 | 0.074 | 0.169 | 0.096 | 0.198 | 0.199 | 0.301 | 0.268 | 0.379 | 0.140 | 0.252 | 0.386 | 0.461 |
| | 24 | **0.059** | **0.139** | 0.185 | 0.290 | 0.074 | 0.173 | 0.065 | 0.155 | 0.201 | 0.301 | 0.081 | 0.178 | 0.105 | 0.209 | 0.225 | 0.325 | 0.256 | 0.370 | 0.174 | 0.289 | 0.708 | 0.644 |
| | 72 | **0.071** | **0.160** | 0.183 | 0.291 | 0.081 | 0.187 | 0.077 | 0.170 | 0.222 | 0.316 | 0.077 | 0.178 | 0.109 | 0.215 | 0.317 | 0.338 | 0.285 | 0.396 | 0.202 | 0.321 | 0.510 | 0.552 |
| | 144 | **0.079** | **0.171** | 0.184 | 0.292 | 0.085 | 0.193 | 0.085 | 0.181 | 0.229 | 0.322 | 0.088 | 0.192 | 0.113 | 0.220 | 0.378 | 0.425 | 0.372 | 0.468 | 0.204 | 0.322 | 0.468 | 0.528 |
| | Avg | **0.065** | **0.150** | 0.180 | 0.286 | 0.077 | 0.179 | 0.071 | 0.162 | 0.216 | 0.311 | 0.080 | 0.179 | 0.106 | 0.210 | 0.280 | 0.360 | 0.295 | 0.403 | 0.180 | 0.296 | 0.518 | 0.547 |
| Payment | 12 | **0.050** | **0.121** | 0.123 | 0.230 | 0.065 | 0.156 | 0.152 | 0.145 | 0.184 | 0.265 | 0.094 | 0.171 | 0.090 | 0.180 | 0.164 | 0.249 | 0.272 | 0.349 | 0.129 | 0.229 | 0.382 | 0.437 |
| | 24 | **0.062** | **0.135** | 0.144 | 0.249 | 0.077 | 0.167 | 0.178 | 0.165 | 0.183 | 0.266 | 0.099 | 0.178 | 0.108 | 0.196 | 0.216 | 0.280 | 0.265 | 0.343 | 0.157 | 0.266 | 0.345 | 0.412 |
| | 72 | **0.082** | **0.155** | 0.151 | 0.251 | 0.094 | 0.184 | 0.236 | 0.193 | 0.226 | 0.287 | 0.111 | 0.189 | 0.129 | 0.209 | 0.360 | 0.370 | 0.284 | 0.360 | 0.183 | 0.291 | 0.437 | 0.471 |
| | 144 | **0.093** | **0.166** | 0.154 | 0.251 | 0.101 | 0.190 | 0.260 | 0.214 | 0.240 | 0.294 | 0.115 | 0.189 | 0.138 | 0.215 | 0.410 | 0.391 | 0.379 | 0.441 | 0.194 | 0.296 | 0.501 | 0.518 |
| | Avg | **0.072** | **0.144** | 0.143 | 0.245 | 0.084 | 0.174 | 0.207 | 0.179 | 0.208 | 0.278 | 0.105 | 0.182 | 0.116 | 0.200 | 0.288 | 0.322 | 0.300 | 0.373 | 0.166 | 0.271 | 0.417 | 0.460 |
| Customer | 12 | **0.065** | **0.129** | 0.191 | 0.247 | 0.091 | 0.160 | 0.243 | 0.156 | 0.267 | 0.289 | 0.123 | 0.180 | 0.143 | 0.195 | 0.310 | 0.326 | 0.309 | 0.366 | 0.175 | 0.243 | 0.640 | 0.580 |
| | 24 | **0.078** | **0.141** | 0.214 | 0.264 | 0.107 | 0.173 | 0.293 | 0.177 | 0.267 | 0.291 | 0.130 | 0.183 | 0.170 | 0.212 | 0.338 | 0.344 | 0.313 | 0.369 | 0.188 | 0.264 | 0.763 | 0.642 |
| | 72 | **0.108** | **0.161** | 0.222 | 0.266 | 0.131 | 0.190 | 0.331 | 0.215 | 0.334 | 0.317 | 0.149 | 0.196 | 0.202 | 0.228 | 0.511 | 0.408 | 0.330 | 0.374 | 0.267 | 0.324 | 0.616 | 0.564 |
| | 144 | **0.126** | **0.172** | 0.227 | 0.268 | 0.141 | 0.195 | 0.368 | 0.226 | 0.363 | 0.332 | 0.166 | 0.206 | 0.222 | 0.239 | 0.687 | 0.461 | 0.450 | 0.456 | 0.336 | 0.373 | 0.658 | 0.586 |
| | Avg | **0.094** | **0.150** | 0.214 | 0.261 | 0.118 | 0.180 | 0.309 | 0.194 | 0.308 | 0.307 | 0.142 | 0.191 | 0.184 | 0.219 | 0.461 | 0.385 | 0.350 | 0.391 | 0.242 | 0.301 | 0.669 | 0.593 |
| 1st Count | | **28** | **27** | 0 | 0 | 0 | 0 | 0 | 3 | 0 | 0 | 2 | 0 | 0 | 0 | 0 | 0 | 0 | 0 | 0 | 0 | 0 | 0 |

# G   DISCUSSIONS AND FURTHER IMPROVEMENT

## G.1   DISCUSSIONS ON ARCHITECTURE-FREE METHODS

Channel Independence (CI) (Nie et al., 2023), regarding variates of time series independently and adopting the shared backbone, have gained increasing popularity in forecasting with performance promotions as an architecture-free method. Recent works (Han et al., 2023; Li et al., 2023) found that while Channel Dependence (CD) benefits from a higher capacity ideally, CI can greatly boost the performance because of sample scarcity, since most of the current forecasting benchmarks are not large enough. We think it is essential to make variates independent, especially when there are potential risks of embedding as mentioned in Appendix E.3, inducing the ideal model capacity of CD limited by the excessively localized receptive field. However, the essence of CI, regarding multivariate time series univariately, can lead to time-consuming training and inference and become an obstacle to scalability. Still, multivariate correlations can not be explicitly utilized. Perpendicular to these works, iTransformer repurposes an architecture with the native Transformer modules to tackle the issues.

RevIN (Kim et al., 2021) and Stationarization (Liu et al., 2022b) have been widely applied for the distribution shift (non-stationarity) as architecture-free techniques. These works strive to reveal the temporal dependency better. This is accomplished by layer normalization in iTransformer and still leaves further improvement for us to tackle the distribution shift.

## G.2   DISCUSSIONS ON LINEAR FORECASTERS

Linear forecasters have natural advantages in modeling temporal dependencies. The dense weighting (Zeng et al., 2023; Li et al., 2023) can reveal measurement-free relationships among the time points of the same variate. More advanced linear forecasters focus on structural point-wise modeling (Oreshkin et al., 2019; Liu et al., 2022a; 2023). By contrast, iTransformer is particularly good at forecasting high-dimensional time series (numerous variates with complicated correlations, which can be common and realistic for practitioners in real forecasting applications). For variate correlating, the embedding keeps the variate independent and the attention module can be applied to dig it out. Under univariate scenarios, iTransformer actually becomes a stackable linear forecaster (attention degradation), which leaves further enhancement to exploit the temporal dependency better.

## G.3   DISCUSSIONS ON TRANSFORMERS

We emphasize that iTransformer actually proposes a new perspective to think about the multivariate time series modality, specifically, how to consider the variates and the tokenization. We list several representatives in Figure 19. Transformer treats time series as the natural language but the time-aligned embedding may bring about risks in multi-dimensional series. The problem can be alleviated by expanding the receptive field. Although it is believed that Patching (Zhang & Yan, 2023; Nie et al., 2023) can be more fine-grained, it also brings higher computational complexity and the potential interaction noise between time-unaligned patches. If the current embedding (implemented by MLP) is enhanced with more inductive bias (such as TCN), it may handle more robust cases with the variate token paradigm and enjoy the flexibility of Transformer with changeable numbers of tokens.

We believe the capability and scalability of Transformer have stood the test by extensive fields, but there is still improvement room to elaborately design components based on the inverted architecture, such as efficient attention for multivariate correlation, structural temporal dependency modeling under distribution shift, fine-grained variate tokenization and well-designed embedding mechanisms.

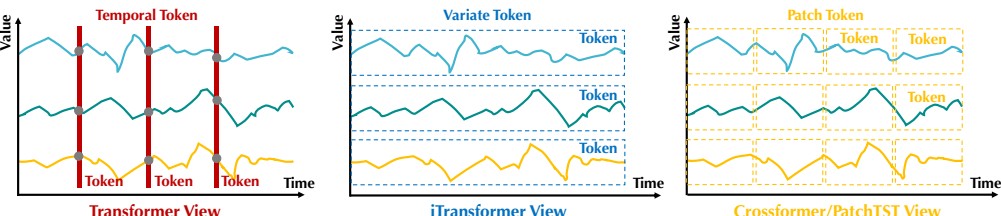

Figure 19: Tokenizations for multivariate time series modality of representative Transformers.

