# OpenReview forum: "iTransformer: Inverted Transformers Are Effective for Time Series Forecasting"
_ICLR.cc/2024/Conference — ICLR 2024 spotlight_

### Official Review · Reviewer_b5RW · 2023-10-27

**Soundness:** 3 good
**Presentation:** 3 good
**Contribution:** 3 good
**Rating:** 8
**Confidence:** 4

**Summary:**

The paper addresses the problem of time series forecasting.
The authors propose a model that embeds univariate
channels as a whole and then uses attention between
embedded channels. In experiments on several datasets
they show that their model outperforms current models
and established new state of the art results.

**Strengths:**

s1. very simple idea and model.
s2. very good results, establishing new state of the art results.
s3. interesting additional finding that longer lookback windows
  now are mostly beneficial (fig. 6).

**Weaknesses:**

w1. experiments only on selected datasets compared to some
  major baselines.
w2. experimental results for close baseline PatchTST varies from
  published results.
w3. no standard deviations reported.

**Questions:**

The paper proposes a very simple idea, but the experiments
to the best of my knowledge are establishing a new state of the
art, making it an important contribution like an indepth study.
I also liked the ablation study with growing observation horizons,
as they now are more plausible than in the related work: longer
observation horizons usually pay off for your model.

Some points should be discussed:
w1. experiments only on selected datasets compared to some
  major baselines.
- You do not report on datasets Exchange, ILI and ETTm1, ETTm2,
  ETTh1 and ETTh2, different from the experiments reported in TimesNet.
  This way it is hard to see if the proposed method really outperforms
  the baselines consistently or just on the selected datasets.

w2. experimental results for close baseline PatchTST varies from
  published results.
- PatchTST consistently reports better results, e.g., for Electricity
  with horizon 96 they report an MSE of 0.129, you report 0.195.
  Where does the difference come from?

w3. no standard deviations reported.
- Standard deviations will help to assess which differences might be
  significant and which spurious.

Some minor language issues:
- abstract, "However, Transformer is challenged": missing "a".
- p. 2 "irrationality": sounds a little bit too strong to me.

---

> ### Author Response · Authors · 2023-11-18
> **Response to Reviewer b5RW**
>
> Many thanks to Reviewer b5RW for providing thorough detailed comments.
>
> **Q1:** Robustness and completeness of the experiments.
>
> Thanks for your scientific rigor. We provide the additional experiments in our revised paper:
>
>   * Report the standard deviations of iTransformer performance in $\underline{\text{Table 5 of the revised paper}}$ with five random seeds,  **exhibiting stable performance**.
>   * Update the main results in $\underline{\text{Figure 1 and Table 1 of the revised paper}}$. We report the averaged results from four subsets of ETT and PEMS. We also add the Exchange dataset. The detailed results are updated in  $\underline{\text{Table 9 and 10 of the revised paper}}$.
>   * Full results of the inverted Transformers on more datasets in $\underline{\text{Appendix E and F of the revised paper}}$ to further support the conclusions of our model analysis.
>
> **Q2:** About the different PatchTST results from its original paper.
>
> We carefully read through the forecasting settings of the PatchTST paper and its official repo. There's a lot of difference here:
>
>   * **Enlarged lookback window**: PatchTST paper adopts tunable lookback length in $\{336, 512\}$, while ours adopts the unified length of $96$, following the unified long-term forecasting protocol of TimesNet.
>   * **More epochs to train**: PatchTST paper trains the PatchTST model with $100$ epochs, while we train all models with $10$ epochs.
>   * **Learning rate**: PatchTST paper adapts a carefully designed learning rate strategy.
>
> **Q3:** About the writing issues.
>
> Thanks for your valuable suggestions. We conduct the following revisions of our paper:
>
>   * Abstract: "Transformer is challenged" -> "Transformers are challenged".
>   * Introduction: "Concerning irrationality of embedding multivariate points" -> "Concerning the potential risks of embedding multivariate points".

---

### Official Review · Reviewer_HTy3 · 2023-10-30

**Soundness:** 3 good
**Presentation:** 3 good
**Contribution:** 3 good
**Rating:** 8
**Confidence:** 4

**Summary:**

In this paper, Authors propose to investigate why Transformer-based models do not seem to be as efficient as Linear-based models for Multivariate Time Series Forecasting (MTSF), while they are predominant in other AI domains. They suggest that the way Transformer is implemented for MTSF is inappropriate. To better benefit from Transformer architecture, they propose to tokenize dataset based on variate and not on timestep. It ends up modifying the input and the FFN. Their proposal also uses only an encoder compared to the vanilla Transformer architecture.

Authors conduct extensive experiments with several Linear-based and Transformer-based baselines to determine the performance of their proposal. These experiments are performed over 6 usual MTSF datasets along with 6 market-based datasets. Their proposal, iTransformer, appears to achieve best predictions for all datasets despite the selected prediction horizon.

**Strengths:**

In my opinion, this paper will have a high impact on MTSF research and community. They back-up their proposal with extensive experiments on several datasets and compared to multiple baselines.
The results and ablation study are discussed even though I feel it could have gone further, but the page limitation was probably the issue.

**Weaknesses:**

The biggest weaknesses in my opinion are first, the writing style, which sometimes is not going to the point (could be improved to better convey the idea and have greater impact) and second that Authors fail to include some important SOTA baselines and properly synthetize/discuss CD/CI/distribution shift comparing all the baselines (including the missing ones) and iTransformer.

**Questions:**

# Paper as is
As far as I am concern, the paper in its current version with some proof-reading and the following revisions can be Accepted.
 * In abstract “the duties of the attention mechanism” should be “the following duties of the […]” to make reading smoother and avoid readers from questioning on what duties means here.
 * The following claim “with potentially unaligned timestamps” need to be referenced or proven. Which datasets have unaligned timestamps, or by unaligned does Authors means that there are delays between values from variates? In other terms, are physical measurements unaligned (which is not the case in the considered dataset)? or do values that correspond to the same “event” appears at different timesteps for different variates (which means that there are delays, which might be the case for Traffic, but for the others, in my opinion, Authors would need to demonstrate it. For instance, in Weather (which are data from the same weather station), it is clearly not the case).
 * In my opinion, it is more appropriate to call Electricity dataset as ECL.
 * Reproducibility
   * Avoid confusion and precise which ETT is used. It looks like it is ETTm according to appendix (15min frequency), but is it m1 or m2? Informer results looks like ETTh though…

   * Also, precise which PEMS you use PEMS-Bay (occupancy ratio or speed in San Francisco Bay?) or other?
 * In my opinion, Figure 1 is focusing too much on iTransformer results. Why not having each axis of the web going from 1 (in the center) to 0? For better fairness in the visualization.
 * Figure 7 (b?) right part is difficult to understand without explanation of x and y axis, and what is the differences between Score Map and Correlations. What is important in this figure? How should reader interpret these lines. Why is it better than usual Transformer?
 * Figure 8 needs label of x axis.
 * “which can be attributed to the extremely fluctuating series of the dataset, and the patching mechanism of PatchTST may lose focus on specific locality to handle rapid fluctuation. By contrast, our proposed method aggregating the whole series variations for series representations can better cope with this situation” I would need proof for such a claim (even in appendix). First, showing this situation and second showing learning weight for iTransformer that shows it cope with the situation.

Proof-read is required for instance:
 * “this goal can be hardy achieved” I guess there is a typo here and Authors was aiming to write hardly?
 * “Soloar-Energy”- > Solar Energy


# Toward a bigger impact

## Additional baselines and larger scope
Nonetheless, I feel that Authors are missing the opportunity to have an even greater impact in the MTSF community (And having my contribution score going from Good to Excellent). Indeed, the proposal is very promising and is on a very important topic, i.e., how to consider variate in MTSF. Are they channel dependent (CD) or channel independent (CI), and if CI are projection perform commonly or individually? However, in my humble opinion, despite doing a good job to present their proposal and results with ablation study and visualization, Authors fail to really position their proposal in the landscape of the above question. It is true that they compare they work to PatchTST, and the CD/CI discussion, but RLinear or RMLP (depending on the dataset) [2] appears to also beat PatchTST. And especially, RLinear with individual projection (one linear layer per variate) similarly to NLinear or DLinear performs better. In addition, RLinear or RMLP use RevIN that was proposed in [1]. The latter show that RevIN helps to handle distribution shift and could be applied to Transformer-based models such as Informer to improve their performance.

Therefore, in my opinion, the paper will have a greater impact if Authors compare their results to these following baselines: revInformer, RLinear and RMLP (and so corresponding papers). And discuss the results and impact of inverse versus CI versus distribution shift handling. This extra step would really be significant for the community in order to have a better understanding of the big picture and what is happening here.

In addition, the abstract and intro emphasis that Transformer superiority is “shaken” by Linear-based models. However, Authors have only few of such Linear models as baselines. Therefore, adding RLinear and RMLP, especially if iTransformer can beat them, will emphasis such a claim.

Authors cited [2] in their paper, so they are aware of this work and in my opinion should have included it.

Finally, for the experiments where only P% of the variates are used for training, as the variates are selected randomly, it would be good to perform the experiment with different set of random variates and plot an error plot or box plot. This will further highlight that the set selected randomly is not a specific case. For instance, in Figure 8 (a? left one), we could have for each % of variate the min, max, average among the different sets. Figure 5 could be a boxplot. In addition, Authors should make clear that the set of variates use with iTransformer is the same set used in CI transformer (Figure 5) to avoid any misunderstanding from Reader.

## More results in appendix to ensure proper reproducibility
Furthermore, in order to target greater impact on the community, I would suggest Authors to add results and visualizations for the other datasets (similar to Table2, Figure 5, and following) in appendix. This would help to make sure results showed applied to all datasets. Indeed, Authors mentioned that PEMS is more difficult so the nature/type/characteristics of the dataset may impact the results and it would be important to mention it and discuss it (even though this point might what Authors expect to do as future work by saying “explore iTransformers for extensive time series analysis tasks”).

Especially, I also expect Figure 6 for dataset like Solar energy to not be as good as the others. Because seasonality of Solar energy is high and it strongly depend on the weather, so increasing the lookback window might not be that beneficial.



[1] https://openreview.net/pdf?id=cGDAkQo1C0p

[2] https://arxiv.org/pdf/2305.10721.pdf

---

> ### Author Response · Authors · 2023-11-18
> **Response to Reviewer HTy3 (Part 1)**
>
> We would like to sincerely thank Reviewer HTy3 for providing a detailed review and insightful suggestions.
>
> **Q1:** About the writing and figure revisions.
>
> Thanks for your valuable suggestions. We conduct the following revisions of our paper:
>
>   * Misspelling: hardy -> hardly; Soloar-Energy -> Solar Energy.
>   * Abstract: "inverts the duties of the attention mechanism and the feed-forward network" -> "applies the attention and feed-forward network on the inverted dimensions". The detailed duties are elaborated in $\underline{\text{Section 3.2}}$.
>   * Figure 8: add the label of the x-axis.
>   * Abbreviate the Electricity dataset as ECL.
>
> As for the fairness of $\underline{\text{Figure 1}}$, the results corresponds strictly to $\underline{\text{Table 1}}$. We set the maximum value of the radar as the best result of the benchmark for aesthetic reasons.
>
> **Q2:** Avoid dataset confusion and be precise on ETT and PEMS datasets.
>
> Thanks for your scientific rigor. To avoid confusion, we extensively test the performance of all four ETT subsets and the same four PEMS subsets as the SCINet. **We report the averaged results of these subsets** in $\underline{\text{Figure 1 and Table 1 of the revised paper}}$ to  The detailed dataset descriptions are updated in $\underline{\text{Appendix A.1 of the revised paper}}$.
>
> To be more precise and reproducible, we also **report the standard deviations of iTransformer performance in five runs, indicating stable results**, which are updated in $\underline{\text{Table 5 of the revised paper}}$.
>
> **Q3:** The proof supports the potential risks of unaligned timestamps and measurements on specific data sets.
>
> We provide a detailed analysis of the Traffic dataset in $\underline{\text{Appendix E.3 of the revised paper}}$. We find **there are systematical delays in the road occupancy that each series describes**, because the sensors are installed in different areas (imagine how a traffic jam influences the road occupancy of different areas).
>
> Delay detection is the essential work that the MTSF model should cope with. However, it is because the vanilla Transformer embeds the localized time points that the obvious decay will be fused with noise.
>
> Another support comes from the declined Traffic performance of the second and third rows in $\underline{\text{Table 3}}$, where attention is applied to the tokens of simultaneous time points. Since they do not reflect the same event, the performance can degrade a lot because of meaningless attention maps, unless the model has an enlarged respective field to capture the decay.
>
> As for the unaligned measurements, it is common in time series forecasting, such as organizing together the different meteorological indicators (temperature and rainfall in Weather), or the quantity and proportion of the same observation (ILI dataset).
>
> **Q4:** More detailed explanations about $\underline{\text{Figure 7(b)}}$ about the multivariate correlations.
>
> We newly add detailed explanations about multivariate correlations in $\underline{\text{Appendix E.1 of the revised paper}}$. In a nutshell, the x- and y-axis place the variates, and **the attention map can thus exhibit the Pearson Correlations among variates**. We want to show that the attention maps can reflect the correlations of raw series (comparing the subplots of the same column), benefiting Transformer with enhanced interpretability. Besides, the correlations, which vary from the lookback window to forecast window, will be reflected by the attention maps that change accordingly (comparing the subplots of the same row).
>
> **Q5:** The proof supports the fluctuation of PEMS.
>
> We provide PEMS prediction visualization in $\underline{\text{Appendix E.2 of the revised paper}}$. **It shows that PEMS has more fluctuating series variations compared with others**. We highlight "PatchTST may lose focus on specific locality to handle rapid fluctuation" for that the patch length should be adaptively tuned for the series (such as frequency). Instead, iTransformer embeds the whole series as a token to describe the underlying process globally.

---

> ### Author Response · Authors · 2023-11-18
> **Response to Reviewer HTy3 (Part 2)**
>
> **Q6:** Discussions on the Channal-Independence/Dependence, Instance Normalization, and the linear models.
>
> Thanks for your valuable suggestion. **We newly added RLinear to our baseline**, which is very competitive on several datasets, such as ETT and Weather.
>
> As the reviewer insightfully mentioned the CI/CD and RevIN[1], it is important to discuss on these techiques:
>
>   - Previous works[2]\[3] have discussed the CI/CD tradeoff on model capacity and robustness. For example, CI can greatly increase the performance when data is scarce, while CD benefits from a higher capacity ideally. In the context of our work, **we think it is essential to make the variates independent**, especially when we know there are potential risks of delayed series and loosely organized measurements. Still, no matter CI/CD, **multivariate correlations** can not be explicitly utilized. Perpendicular to these works, iTransformer repurposes an architecture with the native Transformer modules to cope with it.
>   - RevIN or Series Stationarization[4] has been widely applied for the distribution shift (or non-stationarity) for real-world time series. They can boost the performance of both linear models and Transformers. There are Transformers in our baseline already equipped with this technique (PatchTST and Stationary), and we newly include the linear one, RLinear. These works strive for modeling **temporal dependencies**. In iTransformer, it is modeled naturally by the FFN and layernorm, and still leaves further improvement for us to tackle the distribution shift.
>   - Compared with linear forecasters, **iTransformer is good at forecasting high-dimensional time series** (that is, numerous variates, and complicated correlations). Under the univariate forecasting scenarios, iTransformer becomes in fact a stackable linear forecaster (FFN with the laynorm). For variate correlating, iTransformer keeps the variate independent and the attention module is on duty.
>
> [1] RevIN: Reversible Instance Normalization For Accurate Time-series Forecasting Against Distribution Shift.
>
> [2] The Capacity and Robustness Trade-off: Revisiting the Channel Independent Strategy for Multivariate Time Series Forecasting.
>
> [3] Is Channel Independent strategy optimal for Time Series Forecasting?
>
> [4] Non-stationary Transformers: Exploring the Stationarity in Time Series Forecasting.
>
> **Q7:** Complete experiments to enhance the robustness of the results and reproducibility.
>
> We extensively include the following experiments to our paper:
>
>   * To support $\underline{\text{Table 2}}$: promotion of the inverted Transformer on all datasets in $\underline{\text{Appendix F.1 of the revised paper}}$.
>
>   * To support $\underline{\text{Figure 5}}$: additional results of PEMS to show iTransformer can generalize on unseen variates in $\underline{\text{Appendix F.3 of the revised paper}}$.
>
> Additional results of increasing lookback on Solar-Energy: increasing the lookback length lead to an U-curve of performance (change point in $336$ and $720$) on this dataset. We also provide the explanations in $\underline{\text{Q4 of Reviewer yopU}}$.
>
> **Q8:** Clarify the protocol for the experiments trained with partial variates.
>
> Different from the efficiency training strategy, which randomly chooses part of the variates in each batch, there is no randomness in variate selection in $\underline{\text{Figure 5}}$. We conventionally take **the first $20\%$ variate** of each dataset for training and use all for inference. So **CI-Transformer and iTransformer use the same set of variates during the whole process of training**. We have revised the part with a more rigorous description.
>
> We also elaborate more on the the efficiency training strategy with more datasets added, and comprehensively test the model efficiency and performance in $\underline{\text{Appendix D of the revised paper}}$.

---

> ### Comment · Reviewer_HTy3 · 2023-11-21
>
> Thank you for the revision of the paper and replies to my review.
> As I already said in my review, the paper has enough material to be accepted and Authors applied some of my comment as well as other reviewers' comments, which is, in my humble opinion, "bullet-proofing" the paper even more.
>
> But here are few comments based on the new version:
>
> ## Data unaligned
> I would just notify Authors that in the paper they mentioned dataset with "unaligned timestamps". However, in all the dataset Authors are using, timestamps are not unaligned. It is just the effect of specific events that is delayed, and it is only due to spatial nature of some dataset. Therefore, they should not say there are dataset with unaligned timestamp (because aligning variates for these specific events will results in misalignment for other timestamps) but rather talk about delayed or unaligned event.
>
> ## Market Dataset
> FYI, Authors did not provide the results of RLinear.
>
> ## Protocol for experiments with partial variates
> > We conventionally take the first 20% variates of each dataset [...]
>
> Then, my comment still stands. What would have been the results if Authors had taken the last 20% variates? or any other set of 20% variates.
> There is a risk that taking the first 20% variates is a specific scenario for which the transfer works correctly. Therefore, to be rigorous it would have been better to try various configurations and check if the performances vary a lot or if it is stable.
>
> ## Regarding CD/CI/DistributionShift discussion
> >  Discussions on the Channel-Independence/Dependence, Instance Normalization, and the linear models.
>
> In my opinion, the comments of the Authors (and their opinion on this important topic) should be in the paper, even a condensed version. However, right now I can't see it in the revised version. But I do understand that the paper "format" (and page limitation) prevents to fully discuss and further investigate this topic.
>
> Anyway, I'm convinced that the proposal of iTransformer will be an important milestone for Multivariate Time Series Forecasting, therefore no problem on my side for the new version of the paper.

---

> > ### Author Response · Authors · 2023-11-22
> > **Additional Response to Reviewer HTy3 (Part 1)**
> >
> > Thanks a lot for your valuable comments, which helped us greatly in the rebuttal and paper revision. We'd like to provide more elaborations on the point of your dedicated response.
> >
> > **Q1:** Correct the clarification of "datasets with unaligned timestamps".
> >
> > Thank you very much for the correction. We notice that "datasets with unaligned timestamps" is not a precise expression. No matter whether a dataset is time-aligned or not, there are still risks of embedding multivariate time points of the same timestamp.
> >
> > We will avoid simply attributing the risks to "datasets with unaligned timestamps" and elaborate more on the detailed scenarios in the final version of the paper, such as delayed events (as you instructively point out), irregular (unevenly spaced) time series, systematical delay of monitors, and the time interval of generating and recording the time series. It can be beneficial to remark on these scenarios with reflections on the Transformer embedding for time series and understand why an enlarged receptive field (Variates as Tokens or Patching) works better for the time series modality.
> >
> > **Q2:** RLinear results on Market Dataset.
> >
> > As per your suggestion, we newly include the RLinear results of the Market baseline in $\underline{\text{Table 11 of the revised paper}}$. It can be observed that RLinear (as well as other linear forecasters) is not particularly good on this dataset. The market dataset, which records the server load of an online transaction platform, generally includes hundreds of variates (from 285 to 795). In detail, each variable represents the monitored values of a service interface of a kind, and the service can be further grouped into refined categories. So it is important to exploit the multivariate correlations in such a challenging task.

---

> > ### Author Response · Authors · 2023-11-22
> > **Additional Response to Reviewer HTy3 (Part 2)**
> >
> > **Q3:** Try various configurations for the experiments with partial variates.
> >
> > Thanks for your scientific rigor. We newly conduct $60$ experiments on partial variates, which list the variate generalization results of all five folds with 20% partial variates, as well as the average (**Avg**) and standard deviation of the five runs. We list the comprehensive generalization error as **Comp** by calculating the increasing ratio of MSE from **Full** (full-variate training) to **Avg**.
> >
> > It seems that taking the first 20% variates instead of the last 20% results in slightly better transfer on ECL. But the conclusion can still hold that our model can feasibly generalize on unseen variates with a small increase in forecasting error on Traffic and Solar Energy.
> >
> > As you rigorously point out, it is important to perform the experiment with different sets of variates and plot the averaged results. Due to the limited time, we are unable to complete Channel-Indpendent experiments since it is really time-consuming on the dataset with numerous variates. We promise to update the Figures with the averaged results from all folders in the final version.
> >
> > | MSE (iTransformer) | ECL                  | Traffic             | Solar Energy         |
> > | ------------------ | -------------------- | ------------------- | -------------------- |
> > | Full (0-100%)      | 0.162                | 0.443               | 0.202                |
> > | 0-20%              | 0.192                | 0.444               | 0.214                |
> > | 20%-40%            | 0.210                | 0.448               | 0.210                |
> > | 40%-60%            | 0.210                | 0.448               | 0.242                |
> > | 60%-80%            | 0.222                | 0.445               | 0.239                |
> > | 80%-100%           | 0.222                | 0.445               | 0.225                |
> > | Avg ± Std  (Comp)  | 0.211±0.011 (+30.2%) | 0.445±0.001 (+0.6%) | 0.226±0.013 (+11.8%) |
> >
> >
> > | MSE (iInformer)   | ECL                  | Traffic             | Solar Energy         |
> > | ----------------- | -------------------- | ------------------- | -------------------- |
> > | Full (0-100%)     | 0.169                | 0.459               | 0.200                |
> > | 0-20%             | 0.204                | 0.466               | 0.233                |
> > | 20%-40%           | 0.211                | 0.465               | 0.228                |
> > | 40%-60%           | 0.218                | 0.465               | 0.237                |
> > | 60%-80%           | 0.217                | 0.477               | 0.238                |
> > | 80%-100%          | 0.215                | 0.468               | 0.230                |
> > | Avg ± Std  (Comp) | 0.213±0.005 (+26.0%) | 0.468±0.005 (+2.0%) | 0.233±0.004 (+16.4%) |
> >
> >
> > | MSE (iReformer)  | ECL                  | Traffic             | Solar Energy        |
> > | ---------------- | -------------------- | ------------------- | ------------------- |
> > | Full (0-100%)    | 0.170                | 0.459               | 0.198               |
> > | 0-20%            | 0.190                | 0.463               | 0.204               |
> > | 20%-40%          | 0.202                | 0.462               | 0.203               |
> > | 40%-60%          | 0.203                | 0.463               | 0.207               |
> > | 60%-80%          | 0.221                | 0.463               | 0.215               |
> > | 80%-100%         | 0.202                | 0.466               | 0.208               |
> > | Avg ± Std (Comp) | 0.204±0.010 (+20.0%) | 0.463±0.001 (+1.0%) | 0.207±0.004 (+4.8%) |
> >
> >
> > | MSE (iFlowformer) | ECL                  | Traffic             | Solar Energy        |
> > | ----------------- | -------------------- | ------------------- | ------------------- |
> > | Full (0-100%)     | 0.168                | 0.449               | 0.204               |
> > | 0-20%             | 0.191                | 0.456               | 0.214               |
> > | 20%-40%           | 0.208                | 0.449               | 0.217               |
> > | 40%-60%           | 0.234                | 0.454               | 0.229               |
> > | 60%-80%           | 0.217                | 0.450               | 0.222               |
> > | 80%-100%          | 0.234                | 0.448               | 0.229               |
> > | Avg ± Std  (Comp) | 0.216±0.016 (+28.5%) | 0.451±0.003 (+0.4%) | 0.222±0.006 (+8.6%) |
> >
> > **Q4:** Include the discussions of CD/CI/Distribution shift in the paper.
> >
> > Thank you for your recognition of our discussion on such an important topic. We think it is essential to compare these architecture-free techniques addressing the time series properties and go further into bigger questions that how to consider time series variates and tokenize the time series. We will also include the full discussion and further investigation on this topic in our final version.
> >
> >
> >
> > Sincerely thanks again for your timely help and kind dedication.

---

> > > ### Comment · Reviewer_HTy3 · 2023-11-22
> > >
> > > Thank you for these additional experiments and results.
> > > I really think this way your paper will help other researchers advance on TSF with transformer-based architecture. These results open up to interesting questions that I think will be tackled in the future.
> > > I look forward to reading the final version.

---

### Official Review · Reviewer_yopU · 2023-11-01

**Soundness:** 3 good
**Presentation:** 4 excellent
**Contribution:** 3 good
**Rating:** 8
**Confidence:** 3

**Summary:**

This paper proposes a simple variant of transformer for time series forecasting, where the embedding is applied on each time series and the attention is across each variate. The idea is simple and effective. The improved performance is shown on various real-world datasets. The paper is well-written and easy to read. The idea can be viewed as a principal way to be adopted on various transformer-based architectures. The numerical experiments on different architectures and analysis of representations/correlations greatly enhance the importance of this work.

**Strengths:**

The authors propose a principal way to apply the transformer-based model for time series forecasting. The idea is simple and effective, and the effectiveness is demonstrated via extensive experiments and ablation studies.

**Weaknesses:**

The paper is relatively short of explanation/justification about the effectiveness of such an approach.

**Questions:**

1. It would be better to describe the train-validation-test split in experiments, like training in past years and predicting in the next year, as it could be tricky for data pre-processing in time series forecasting and cause data leakage issues.

2. After reading this paper, does the author implicitly assume the heterogeneity of variates is more important than temporal dependency in terms of forecasting?

3. There are some interesting results in Table 3. Could the author comment on the not-so-good performance of the second row (both attention)?

4. A minor one: in the left panel of Figure 6, there is a little jump on the red line from 336 to 720. Any reason why? Do multiple runs help?

---

> ### Author Response · Authors · 2023-11-18
> **Response to Reviewer yopU**
>
> Many thanks to Reviewer yopU for providing a detailed review and insightful questions.
>
> **Q1:** Describe the train-validation-test split in experiments.
>
> We adopt the same train-validation-test split protocol as previous works, where the train, validation, and test datasets are **strictly divided according to chronological order to make sure there are no data leakage issues**. The data processing and split protocols are also included in $\underline{\text{Appendix A.1 of the revised paper}}$.
>
> **Q2:** Do we implicitly assume that variate heterogeneity is more important than temporal dependency for forecasting?
>
> We think they are both important to achieve good MTSF performance. **However, the problem is that the heterogeneity of variates can hardly be considered in the vanilla Transformer**. After embedding, the variates are projected into the channels of embedding. It ignores the problem of inconsistent physical measurements and can fail to maintain the independence of variates, let alone capture and utilize the multivariate correlation, which is essential for forecasting with numerous variates, and complicated systems driven by the latent physical process (such as meteorological systems).
>
> In addition, even if the FFN and layernorm seem simpler than attention blocks, **they are efficient and competent in learning the temporal dependency of a series**, which can be traced back to statistical forecasters such as ARIMA and Holt-Winter. They also have no problems with inconsistent measurements since they work on the time points of the same variate, and have an enlarged respective field as the whole lookback series can be embedded as the variate token.
>
> **Q3:** The reason for the not-so-good performance when applying attention to both temporal and variate dimensions.
>
> As we observed the same declined performance of Traffic in the third row, the reason can be that **the designs both apply the attention module to the temporal dimension**. In our opinion, capturing temporal dependencies by attention is not a big problem. But it is **based on the fact that the time points of each timestamp are essentially aligned**.
>
> We analyze the Traffic dataset in $\underline{\text{Appendix E.3 of the revised paper}}$, and we find there are potential risks of systematical delay among the road occupancy that each series describes, since the sensors are installed in different areas of the highway.
>
> Consequently, applying attention to the time points of the same timestamp on such a dataset can make the model learn meaningless attention maps, unless the model has an enlarged respective field to capture the decay or causal process. It is one of the reasons that we advocate embedding the whole series as the variate token.
>
> **Q4:** Explanations about the error jump when the input length increases from $336$ to $720$ in $\underline{\text{Figure 6}}$.
>
> We have also noticed the problem that our approach embedding a large lookback series simply by MLP can be simple and coarse-grained. As we increase the lookback length by a large step, the input series inevitably contains more nonbeneficial information for future prediction, and the model capacity should also increase accordingly. It is instructive for us to develop fine-grained variate tokenization and well-designed embedding mechanisms.
>
> **Q5:** Explanations about the effectiveness of our approach.
>
> We add new sections to analyze the effects of the inverting:
>
>   * Analysis of multivariate correlations in $\underline{\text{Appendix E.1 of the revised paper}}$.
>   * Potential risks of the vanilla Transformer embedding in $\underline{\text{Appendix E.3 of the revised paper}}$.
>   * Full results of the iTransformers in $\underline{\text{Appendix F of the revised paper}}$ to support that our method can boost the Transformers' performance, and generalize on unseen variates.

---

> > ### Comment · Reviewer_yopU · 2023-11-21
> >
> > I appreciate the authors' effort in answering my questions. I raised my score from 6 to 8, as the authors have addressed most of my concerns.

---

> > > ### Author Response · Authors · 2023-11-22
> > >
> > > We would like to thank Reviewer yopU for providing a detailed valuable review, which helps us a lot in the rebuttal and paper revision.
> > >
> > > Thanks again for your response and raising the score! We will include the analysis in our final version.

---

### Official Review · Reviewer_hg6h · 2023-11-01

**Soundness:** 3 good
**Presentation:** 3 good
**Contribution:** 2 fair
**Rating:** 6
**Confidence:** 4

**Summary:**

This paper explored a new angle to apply Transformer model to the multivariate time-series forecasting problem. Without the modification of the original transformer component, the proposed iTransformer inverted the duties of the self-attention mechanism and the feed-forward network. In iTransformer, the feed-forward network was used for series encoding, while the self-attention mechanism captured the correlation among different variates. The authors conducted experiments on six real-world datasets to evaluate the proposed model.

**Strengths:**

1.	This paper provided a simple and effective inverted view to improve transformer-based multivariate time-series forecasters.

2.	Compared with the previous use of Transformer structure (without invert), the iTransformer showed some advantages, including better generalization on unseen variates, and the desired performance improvement over enlarged historical information.

3.	Extensive experiments on different multivariate time-series forecasting tasks were conducted for evaluation. The author compared the proposed model with various baselines, along with a comprehensive modal analysis.

**Weaknesses:**

1.	According to Table 3 and Table 7, most of the result values are relatively small. This suggests that some marginal improvement may be susceptible to random factors (e.g., iTransformer v.s. PatchTST on ETT and Weather dataset, iTransformer v.s. SCINet on PEMS dataset). Therefore, I recommend reporting the standard deviation under different random runs and adding a significance test to provide further insights.


2.	Although the proposed efficient training strategy can reduce the required memory, it would still be better to compare its efficiency with linear models, since recent studies have indicated their advantages in both performance and efficiency.

**Questions:**

1.	Can you please explain why the TiDE results are so different from those reported in their original paper?

---

> ### Author Response · Authors · 2023-11-18
> **Response to Reviewer hg6h**
>
> Many thanks to Reviewer hg6h for providing thorough insightful comments.
>
> **Q1:** The standard deviation of the results and the significance tests.
>
> Thanks for your scientific rigor. We repeat each experiment five times with different random seeds. The standard deviations of iTransformer performance are updated in $\underline{\text{Table 5 of the revised paper}}$, which shows **the performance is stable**.
>
> We also conducted the significance tests to compare iTransformer and the mentioned previous SOTA. We include all subsets of ETT and PEMS. The performance is also averaged from four prediction lengths with each containing five runs of random seeds. The standard deviations and test statistics are listed below, showing that **the performance is on par with the previous SOTA PatchTST and SCINet within the margin of error**.
>
> | MSE     | iTransformer | Previous SOTA | test statistic |
> | ------- | ------------ | ------------- | -------------- |
> | ETT     | 0.383±0.001  | 0.381±0.001   | 2.124          |
> | Weather | 0.258±0.001  | 0.259±0.001   | -2.913         |
> | PEMS    | 0.119±0.001  | 0.121±0.002   | -0.419         |
>
> To our best knowledge, the benchmark of time series forecasting has been excavated for a long time. And PatchTST and SCINet as the SOTA on specific datasets has surpassed concurrent works by a large margin, **iTransformer can be on par with on their advantaged datasets while further go ahead on other chanllenging datasets** (such as ECL and Traffic), and thus achieves the comprehensive SOTA.
>
> **Q2:** Compare the model efficiency with linear models.
>
> We newly measure the model efficiency in $\underline{\text{Figure 10 of the revised paper}}$, which comprehensively compares the training speed, memory footprint, and performance of the following models: iTransformer, iTransformer with our efficient training strategy and iTransformer with an efficient attention module; linear forecasters: DLinear and TiDE; Transformers: Transformer, PatchTST, and Crossformer. We have the following observations:
>
>   * The efficiency of iTransformer exceeds other Transformers in datasets with a small number of variates (such as Weather with $21$ variates). In datasets with numerous variates (such as Traffic with $862$ variates), the memory footprints are basically the same, but iTransformer can be trained faster.
>   * **iTransformer achieves particularly better performance on the dataset with numerous variates**, since the multivariate correlations can be explicitly utilized in our model.
>   * By adopting an efficient attention module or our proposed efficient training strategy on partial variates, **iTransformer can enjoy the same level of speed and memory footprint as linear forecasters**.
>
> **Q3:** About the different TiDE results from its original paper.
>
> We carefully read through the forecasting settings of the TiDE paper and its official repo. The differences are listed here:
>
>   * **Enlarged lookback window**: TiDE paper adopts the lookback length of $720$, while ours uses the length of $96$, following the unified long-term forecasting protocol of TimesNet.
>
>   * **More epochs to train**: TiDE paper trains the TiDE model with $100$ Epochs, while we reproduce the results of all models with $10$ epochs.
>
> We try our best to keep hyperparameters comparable for the fairness of experiments, such as adopting the same hidden dimension and the number of layers. **We also compare their performance under the same lookback length of TiDE, where iTransformer still achieves better results**, since our model can also benefit from an enlarged lookback window.
>
> | MSE          | ECL       | ETTh2     | Exchange  | Traffic   | Weather   | Solar-Energy | PEMS03    |
> | ------------ | :-------- | :-------- | :-------- | :-------- | :-------- | :----------- | :-------- |
> | TiDE         | 0.160     | 0.309     | 0.092     | 0.445     | 0.171     | 0.303        | 0.216     |
> | iTransformer | **0.129** | **0.301** | **0.086** | **0.349** | **0.158** | **0.191**    | **0.071** |

---

> ### Author Response · Authors · 2023-11-22
> **Looking forward to your reply**
>
> Dear Reviewer hg6h,
>
> Thanks for your valuable and rigorous review, which has inspired us to improve our paper further substantially. Following your suggestions, we have enhanced the robustness of experiments with the standard deviations of all datasets reported, newly tested the model efficiency comprehensively, and compared the training protocol with TiDE.
>
> During the rebuttal, we received insightful reviews and valuable comments to improve our paper further. $\underline{\text{Reviewer HTy3}}$ highlighted our contributions and gave high comments. And $\underline{\text{Reviewers yopU}}$ raised the score as we have addressed the concerns. We hope that this new version can address your concerns to your satisfaction and notice our contributions. We eagerly await your reply and are happy to answer any further questions. We kindly remind you that the reviewer-author discussion phase will end soon. After that, we may not have a chance to respond to your comments.
>
> Thanks again for your valuable review. Looking forward to your reply.
>
> Authors

---

### Author Response · Authors · 2023-11-18
**Summary of Revisions**

We sincerely thank all the reviewers for their insightful reviews and valuable comments, which are instructive for us to improve our paper further.

The reviewers generally held positive opinions of our paper, in that the proposed method is "**simple and effective**", "**a new angle**", "**has a high impact on MTSF community**", and "**can be viewed as a principal way to be adopted on various transformer-based architectures**", this paper is "**well-written**" and the experiments are "**extensive**", "**comprehensive**" and "**significant**".

The reviewers also raised insightful and constructive concerns. We made every effort to address all the concerns by providing sufficient evidence and requested results. Here is the summary of the major revisions:

1. **Robustness of experiments**: We provide the standard deviations and hyperparameter sensitivity of the proposed model. We validate the conclusions on more datasets to support that our method can boost the performance of Transformers, and generalize on unseen variates.
2. **Additional baselines and datasets**: We include all subsets of ETT and PEMS as well as Exchange to comprehensively evaluate the performance. We also newly include the concurrent and competitive linear model into our baseline.
3. **Clarification of experiment protocols**: We clarify the data processing, dataset descriptions, forecasting settings, and the fairness of the protocol for the experiments trained with partial variates.
4. **More model analysis and cases**: We provide the model efficiency analysis, where we compare Transformers, the inverted version, and linear models. We also present the attention visualization and the risks of the previous Transformer embedding with more intuitive cases.
5. **Polished writings**: We conduct detailed proofreading and revisions with helpful suggestions from the reviewers. We try to highlight more on the peculiarities and broad impact of our work.

All updates are highlighted in blue. Compared with the first submissions, $\underline{\text{the revised paper}}$ has **additional 5 pages**.

The valuable suggestions from reviewers are very helpful for us to revise the paper to a better shape. We'd be very happy to answer any further questions.

Looking forward to the reviewer's feedback.

---

### Meta-Review · Area_Chair_p2At · 2023-12-05

**Metareview:**

This paper presents inverted transformers for time series forecasting. The insightful reflection inspires some modifications to the standard transformer architecture, leading to strong experiment results when compared with SOTA methods in real-world applications. We thank the authors for responding to the comments of the reviewers and revising their paper accordingly. Although the paper in its current form can be published, the authors are encouraged to consider all the comments and suggestions of the reviewers to make further revisions if necessary before publication.

**Justification For Why Not Higher Score:**

Although the paper is good, it is not good enough to be accepted as an oral paper.

**Justification For Why Not Lower Score:**

It could be accepted as a poster paper if there are many stronger papers. However, this paper has some interesting ideas of general interest. It would be good to consider it for spotlight presentation to bring it to a broader audience.

---

### Decision · Program_Chairs · 2024-01-16

Accept (spotlight)